# BindWeave: Subject-Consistent Video Generation via Cross-Modal Integration

**Zhaoyang Li**[1,2]    **Dongjun Qian**[2]    **Kai Su**[2*]    **Qishuai Diao**[2]    **Xiangyang Xia**[2]
**Chang Liu**[2]    **Wenfei Yang**[1]    **Tianzhu Zhang**[1,3*]    **Zehuan Yuan**[2]
[1]University of Science and Technology of China    [2]ByteDance
[3]National Key Laboratory of Deep Space Exploration, Deep Space Exploration Laboratory

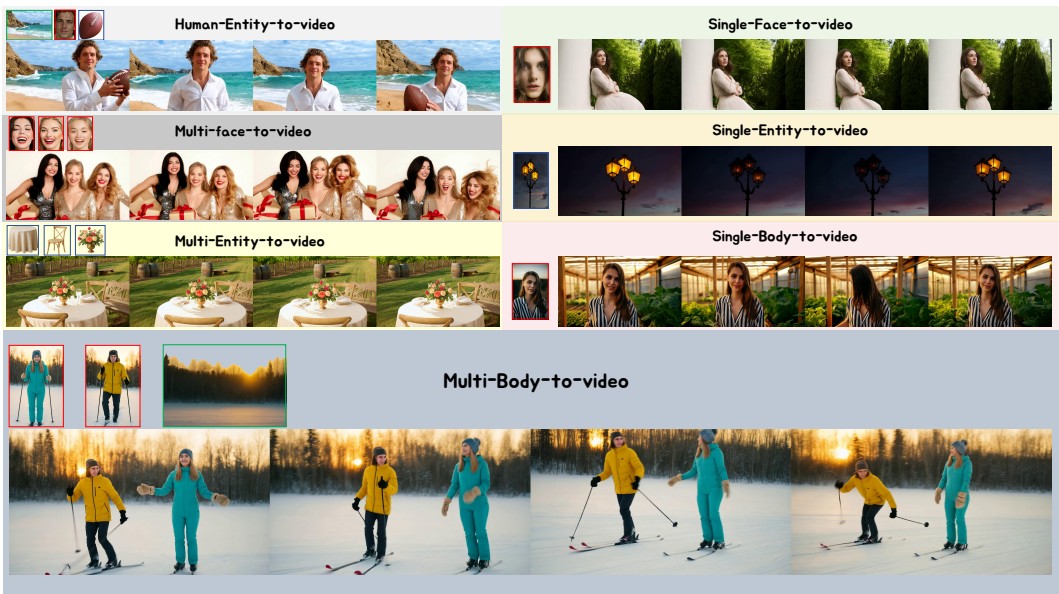

Figure 1: Examples of subject-to-video generation results of our proposed BindWeave, demonstrating its ability to produce high-fidelity, subject-consistent videos across a broad spectrum of scenarios from single-subject inputs to complex multi-subject compositions.

## Abstract

Diffusion Transformer has shown remarkable abilities in generating high-fidelity videos, delivering visually coherent frames and rich details over extended durations. However, existing video generation models still fall short in subject-consistent video generation due to an inherent difficulty in parsing prompts that specify complex spatial relationships, temporal logic, and interactions among multiple subjects. To address this issue, we propose BindWeave, a unified framework that handles a broad range of subject-to-video scenarios from single-subject cases to complex multi-subject scenes with heterogeneous entities. To bind complex prompt semantics to concrete visual subjects, we introduce an MLLM-DiT framework in which a pretrained multimodal large language model performs deep cross-modal reasoning to ground entities and disentangle roles, attributes, and interactions, yielding subject-aware hidden states that condition the diffusion transformer for high-fidelity subject-consistent video generation. Experiments on the OpenS2V benchmark demonstrate that our method achieves superior performance across subject consistency, naturalness, and text relevance in generated videos, outperforming existing open-source and commercial models. Project page: https://lzy-dot.github.io/BindWeave/

---

*Corresponding author

# 1 INTRODUCTION

Recent advances in diffusion models (Ho et al., 2020; Peebles & Xie, 2023; Wang et al., 2025) have enabled significant breakthroughs in video generation (Wan et al., 2025; Yang et al., 2024; Hu et al., 2025; Kong et al.; Polyak et al., 2024; Zheng et al., 2024; Wu et al., 2025), achieving outstanding performance on various tasks ranging from text-to-video (T2V) (HaCohen et al., 2024; Wan et al., 2025; Chen et al., 2025a) to image-to-video (I2V) (Blattmann et al., 2023; Mao et al., 2025). Foundation models such as Wan (Wan et al., 2025) and HunyuanVideo (Kong et al.) now demonstrate the ability to produce high-fidelity, long-duration, and content-rich videos, showcasing immense technological potential. However, despite these advances in visual quality, their practical utility remains constrained by limited controllability. Specifically, current models struggle to exert precise and stable control over key elements within a video, such as the identity of a specific person, the appearance of an object, or a brand logo. This lack of controllability constitutes a core limitation that significantly impedes deployment in customized applications, including personalized content creation, brand marketing, pre-visualization, and virtual try-on.

To address the above challenges, subject-to-video (S2V) (Liu et al., 2025) has garnered increasing attention. The core objective of S2V is to ensure that one or more subjects within a video maintain high fidelity in their identity and appearance with respect to the given reference images throughout the entire dynamic sequence. This capability directly addresses the controllability shortcomings of existing general-purpose models, making it possible to generate customized videos based on user-provided subjects. To achieve subject-consistent video generation, some existing works (Yuan et al., 2024b; Chen et al., 2025b; Huang et al., 2025; Liu et al., 2025; Jiang et al., 2025) extend a video foundation model to accept multiple reference images as conditioning input. For instance, Phantom (Liu et al., 2025) introduces a dual-branch architecture to separately process the text prompt and reference images, subsequently injecting the resulting features into the attention layers of a Diffusion Transformer (DiT) (Peebles & Xie, 2023) as conditioning. VACE (Jiang et al., 2025) designs a video condition unit to unify inputs (text, image/video references, mask) into a unified format, then inject these context signals via residual blocks to guide video generation.

Despite their promising results, these methods share a common limitation: they rely on a separate-then-fuse shallow information processing paradigm. Specifically, these models typically use separate encoders to extract features from images and text independently, followed by a post-hoc fusion through simple concatenation or cross-attention mechanisms. While this mechanism may suffice for simple instructions involving only appearance preservation, its ability to understand and reason falters when faced with text prompts involving complex interactions, spatial relationships, and temporal logic among multiple subjects. Due to a lack of deep semantic association across the multimodal inputs, the model struggles to accurately parse the instructions, often leading to problems like identity confusion, action misplacement, or attribute blending.

To overcome this bottleneck, we propose BindWeave, a novel framework designed for subject-consistent video generation. To establish the deep cross-modal semantic associations, BindWeave leverages a Multimodal Large Language Model (MLLM) as an intelligent instruction parser to replace the conventional shallow fusion mechanism. Specifically, we first construct a unified, interleaved sequence from reference images and text prompt. This sequence is then processed by a pre-trained MLLM to parse complex spatio-temporal relationships and bind textual commands to their corresponding visual entities. Through this process, the MLLM generates a set of hidden states encoding both the precise identity of each subject and their prescribed interactions. These hidden states then serve as conditioning inputs to our generator, bridging high-level parsing with diffusion-based generation. To provide subject-grounded semantic anchors and further reduce identity drift, we also incorporate CLIP (Radford et al., 2021b) features from the reference images. Accordingly, our DiT (Peebles & Xie, 2023) based generator is jointly conditioned on these hidden states and CLIP features. Together, these conditioning inputs provide comprehensive relational and semantic guidance. To preserve fine-grained appearance details, we augment the video latents during diffusion with VAE (Esser et al., 2021) features extracted from the reference images. This collective conditioning on high-level reasoning, semantic identity, and low-level detail ensures the generation of videos with exceptional fidelity and consistency.

We conduct a comprehensive evaluation of BindWeave on the fine-grained OpenS2V (Yuan et al., 2025) benchmark against a diverse set of existing approaches, including leading open-source meth-

ods and commercial models. The evaluation assesses key aspects such as subject consistency, temporal naturalness, and text-video alignment. Extensive experiments demonstrate that BindWeave achieves state-of-the-art performance, consistently outperforming all competing methods in subject-consistent video generation. Qualitative results, illustrated in Figure 1, further demonstrate the superior quality of the generated samples. These findings highlight BindWeave's effectiveness in subject-consistent video generation and its potential as a high-performing solution for both research and commercial applications.

## 2 RELATED WORK

### 2.1 VIDEO GENERATION MODEL

Diffusion models have enabled remarkable advancements in video generation. Early methods (Singer et al., 2022; Blattmann et al., 2023; Guo et al., 2023) often extended text-to-image models (Rombach et al., 2022) for video generation by incorporating temporal modeling modules. More recently, the Diffusion Transformer (DiT) (Peebles & Xie, 2023) architecture, motivated by its excellent scaling properties, has inspired a new wave of models like Wan (Wan et al., 2025), HunyuanVideo (Kong et al., 2024a), and Goku (Chen et al., 2025a). However, these models focus on general-purpose video generation, and there is still considerable room for improvement in achieving fine-grained control.

### 2.2 SUBJECT-CONSISTENT VIDEO GENERATION

To achieve more fine-grained control, subject-consistent video generation has garnered significant attention. Initial approaches often rely on per-subject optimization, where a pre-trained model is fine-tuned on images of a specific subject, as seen in methods like CustomVideo (Wang et al., 2024) and DisenStudio (Chen et al., 2024). While effective, this instance-specific tuning is computationally expensive and poses challenges for real-time applications. More recent works have shifted towards end-to-end methods that use conditioning networks or adapters to inject identity information during inference, allowing for generalization to new subjects without retraining. These models, such as IDAnimator (He et al., 2024) and ConsisID (Yuan et al., 2024a), initially focused on preserving facial identity. This capability was later extended to arbitrary objects and multiple subjects by works like ConceptMaster (Huang et al., 2025), SkyReels-A2 (Fei et al., 2025),Phantom (Liu et al., 2025), and VACE (Jiang et al., 2025). Despite this progress, significant challenges remain, particularly maintaining distinct identities and modeling complex interactions in multi-subject scenes.

## 3 METHOD

### 3.1 PRELIMINARIES

**Diffusion Transformer Models for Text-to-Video Generation.** Transformer-based text-to-video diffusion models have shown substantial promise for video content generation. Our BindWeave builds upon a Transformer-based latent diffusion architecture that employs a spatio-temporal Variational Autoencoder (VAE) (Wan et al., 2025) to map videos from the pixel level to a compact latent space, where the generative process is performed. Each Transformer block comprises spatiotemporal self-attention, text cross-attention, a time-conditioning MLP, and a feed-forward network (FFN). The cross-attention is conditioned on a text prompt embedding $c_{\text{text}}$ obtained from a T5 encoder $\mathcal{E}_{\text{T5}}$ (Raffel et al., 2020). We employ Rectified Flow (Liu et al., 2022; Esser et al., 2024) to define the diffusion dynamics, which enables stable training via ordinary differential equations (ODEs) while maintaining equivalence to maximum likelihood objectives. In the forward process of training, random noise is add to clean data $z_0$ to generate $z_t = (1-t)z_0 + t\epsilon$, where $\epsilon$ is sampled from a standard normal distribution, $\mathcal{N}(0, I)$, and the time coefficient $t$ is sampled from $[0, 1]$. Accordingly, the learning objective becomes the estimation of ground truth velocity field $v_t = dz_t/dt = \epsilon - z_0$. The network $u_\Theta$ is trained to this end using the Flow Matching loss (Esser et al., 2024):

$$\mathcal{L} = \mathbb{E}_{t,z_0,\epsilon,c_{\text{text}}} \|u_\Theta(z_t, t, c_{\text{text}}) - v_t\|_2^2. \tag{1}$$

**Video Generation with Image Conditioning.** Natural language offers an accessible interface for diffusion-based video synthesis, yet it often under-specifies subject identity and spatial layout. This motivates the incorporation of a reference image to anchor appearance and geometry in text-to-video pipelines Wan et al. (2025); Kong et al. (2024b). For instance, Wan (Wan et al., 2025)injects image information in two ways: first at the input level for fine-grained spatial detail, and second within the cross-attention mechanism for semantic guidance. First, to preserve fine-grained appearance, the reference image $\mathcal{I}_{img}$ is encoded by a VAE into a spatial latent $z_{img}$. This latent is concatenated with the current noisy video latent $\mathbf{x}_t$ along the channel dimension. The combined latent is then patchified and linearly embedded to form the initial sequence of tokens for the DiT block:

$$H_{in} = \text{PatchEmbed}(\text{concat}_c(\mathbf{x}_t, z_{img})). \tag{2}$$

Then, semantic guidance is achieved by injecting multimodal conditioning via cross-attention. A pretrained vision encoder $E_{\text{vision}}$ (e.g., CLIP) processes $\mathcal{I}_{img}$ into semantic tokens $\mathcal{H}_{img}$, while a text encoder provides text tokens $\mathcal{H}_{txt}$. Within each cross-attention layer, queries ($\mathbf{Q}$) are derived from the evolving video tokens, denoted as $H_{vid}$ (where $H_{vid}$ is $H_{in}$ for the first layer). The query, key, and value matrices are computed using dedicated linear projection layers ($\Phi_Q, \Phi_K, \Phi_V$):

$$\mathbf{Q}_{vid} = \Phi_Q(H_{vid}), \quad \mathbf{K}_{img} = \Phi_K(\mathcal{H}_{img}), \quad \mathbf{V}_{img} = \Phi_V(\mathcal{H}_{img}), \tag{3}$$

and similarly for $\mathbf{K}_{txt}, \mathbf{V}_{txt}$ from the text stream. The output of the attention layer fuses these information sources using the standard scaled dot-product attention operator, $\text{Attn}(\cdot)$:

$$H_{out} = H_{vid} + \text{Attn}(\mathbf{Q}_{vid}, \mathbf{K}_{txt}, \mathbf{V}_{txt}) + \gamma\,\text{Attn}(\mathbf{Q}_{vid}, \mathbf{K}_{img}, \mathbf{V}_{img}), \tag{4}$$

where $\gamma$ is a scalar balancing the image and text guidance.

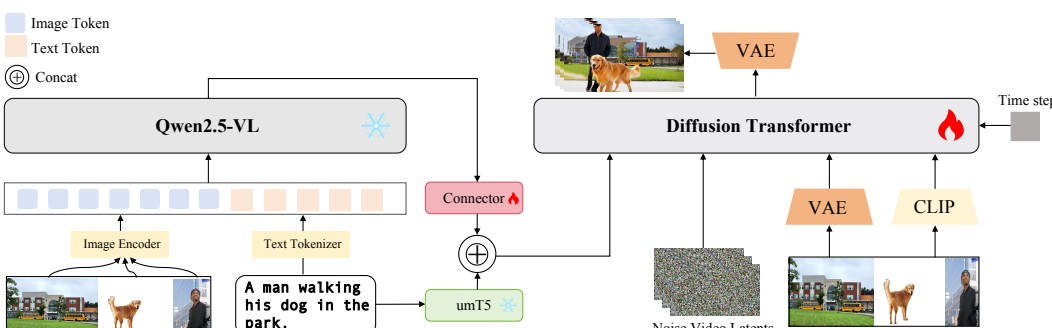

Figure 2: Framework of our method. A multimodal large language model performs cross-modal reasoning to ground entities and disentangle roles, attributes, and interactions from the prompt and optional reference images. The resulting subject-aware signals condition a Diffusion Transformer through cross-attention and lightweight adapters, guiding identity-faithful, relation-consistent, and temporally coherent video generation.

## 3.2 ARCHITECTURE

Our proposed BindWeave is designed to overcome the limitations of shallow fusion paradigms in subject-consistent video generation. The core principle of our approach is to replace shallow, post-hoc fusion with a deep, reasoned understanding of multimodal inputs *before* the generation process begins. To this end, BindWeave first leverages a Multimodal Large Language Model (MLLM) to act as an intelligent instruction parser. The MLLM thus generates a guiding schema, realized as a sequence of hidden states that encodes complex cross-modal semantics and spatio-temporal logic, then meticulously guides a Diffusion Transformer (DiT) throughout the synthesis process. Figure 2 provides a schematic overview of the BindWeave architecture.

## 3.3 INTELLIGENT INSTRUCTION PLANNING VIA MLLM

To effectively foster joint cross-modal learning between the text prompt and reference images, we introduce a unified multimodal parsing strategy. Given a text prompt $\mathcal{T}$ and $K$ user-specified subjects, each with a reference image $I_k$, we constructs a multimodal sequence $\mathcal{X}$ by appending one

image placeholder for each reference image after the text prompt. The MLLM is then provided with this sequence along with the corresponding list of images $\mathcal{I}$:

$$\mathcal{X} = \left[\, \mathcal{T}, \langle \text{img} \rangle_1, \langle \text{img} \rangle_2, \ldots, \langle \text{img} \rangle_K \,\right], \tag{5}$$

$$\mathcal{I} = \left[\, I_1, I_2, \ldots, I_K \,\right], \tag{6}$$

where $\langle \text{img} \rangle_k$ is a special placeholder token that the MLLM internally aligns with the $k$-th image, $I_k$. This unified representation, which preserves the crucial contextual links between textual descriptions and their corresponding visual subjects, is then fed into a pre-trained MLLM. By processing the multimodal inputs $(\mathcal{X}, \mathcal{I})$, the MLLM generates a sequence of hidden states, $H_{\text{mllm}}$, that embodies a high-level reasoning of the scene, effectively binding textual commands to their specific visual identities:

$$H_{\text{mllm}} = \text{MLLM}(\mathcal{X}, \mathcal{I}). \tag{7}$$

To align the feature space between the frozen MLLM and our diffusion model, these hidden states are projected through a trainable lightweight connector, $\mathcal{C}_{\text{proj}}$, to yield a feature-aligned condition $c_{\text{mllm}}$:

$$c_{\text{mllm}} = \mathcal{C}_{\text{proj}}(H_{\text{mllm}}). \tag{8}$$

While this MLLM-derived condition provides invaluable high-level, cross-modal reasoning, we recognize that diffusion models are also highly optimized to interpret fine-grained textual semantics. To provide this complementary signal, we encode the original prompt independently using the T5 text encoder ($\mathcal{E}_{\text{T5}}$) (Raffel et al., 2020) to produce a dedicated textual embedding $c_{\text{text}}$:

$$c_{\text{text}} = \mathcal{E}_{\text{T5}}(\mathcal{T}). \tag{9}$$

We then concatenate these two complementary streams to form our final relational conditioning signal $c_{\text{joint}}$:

$$c_{\text{joint}} = \text{Concat}(c_{\text{mllm}}, c_{\text{text}}). \tag{10}$$

This composite signal thus encapsulates not only the explicit textual commands but also the deep reasoning about subject interactions and spatio-temporal logic, providing a robust foundation for the subsequent generation phase.

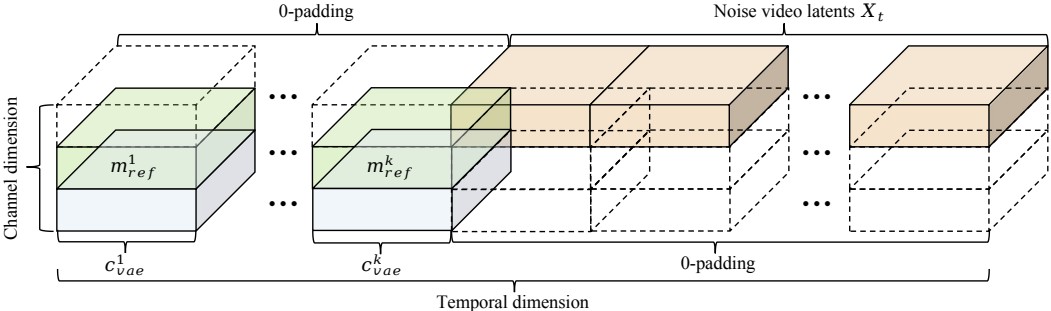

Figure 3: Illustration of our adaptive multi-reference conditioning strategy.

## 3.4 COLLECTIVELY CONDITIONED VIDEO DIFFUSION

In the instruction planning process, we integrate useful semantics into $c_{\text{joint}}$. Now, we need to inject $c_{\text{joint}}$ as a condition into the DiT module to guide video generation. Our generation backbone $u_\Theta$ operating in the latent space of a pre-trained spatio-temporal Variational Autoencoder (VAE). To ensure high-fidelity and consistent video generation, we employ a collective conditioning mechanism that synergistically integrates multiple streams of information. As described in Section 3.1, our collective conditioning mechanism also operates at two synergistic levels: conditioning the spatio-temporal input and the cross-attention mechanism. To maintain fine-grained appearance details from the reference images, we design an adaptive multi-reference conditioning strategy as shown in Figure 3. Specifically, we encode the references into low-level VAE features, denoted as $c_{\text{vae}} = \mathcal{E}_{\text{VAE}}(\{I_{\text{ref}}^i\})$. Since S2V differs from I2V, the reference images are not treated as actual video frames. We first expand the temporal axis of the noisy video latent, padding K additional slots with

zeros: $\tilde{\mathbf{x}}_t = \text{pad}_T(\mathbf{x}_t, K)$. We then place the VAE features of the reference images $c_{\text{vae}}$ onto these K padded time positions (zeros elsewhere), and further concatenate the corresponding binary masks $m_{\text{ref}}$ along the channel dimension to emphasize the subject regions. The final input to the DiT block is obtained via channel-wise concatenation before patch embedding:

$$H_{vid} = \text{PatchEmbed}\big(\text{concat}_c(\tilde{\mathbf{x}}_t, \ \tilde{c}_{\text{vae}}, \ \tilde{m}_{\text{ref}})\big), \qquad (11)$$

where $\tilde{c}_{\text{vae}}$ and $\tilde{m}_{\text{ref}}$ are zero outside the K padded temporal slots and carry the reference conditioning only within those slots. This design preserves the temporal integrity of the original video while injecting fine-grained appearance and subject emphasis through channel-wise conditioning. Then, high-level semantic guidance is injected via the cross-attention layers. This involves two distinct signals: the relational condition $c_{\text{joint}}$ from the MLLM for scene composition, and the CLIP image features $c_{\text{clip}} = \mathcal{E}_{\text{CLIP}}(\{I_{\text{ref}}^i\})$ for subject identity. Within each DiT block, the evolving video tokens $H_{vid}$ generate the queries $\mathbf{Q}_{vid}$. The conditioning signals $c_{\text{joint}}$ and $c_{\text{clip}}$ are projected to form their respective key and value matrices. The final output of the attention layer is a sum of these information streams, extending Equation 4:

$$H_{out} = H_{vid} + \text{Attn}(\mathbf{Q}_{vid}, \mathbf{K}_{\text{joint}}, \mathbf{V}_{\text{joint}}) + \text{Attn}(\mathbf{Q}_{vid}, \mathbf{K}_{\text{clip}}, \mathbf{V}_{\text{clip}}), \qquad (12)$$

where $(\mathbf{K}_{\text{joint}}, \mathbf{V}_{\text{joint}})$ and $(\mathbf{K}_{\text{clip}}, \mathbf{V}_{\text{clip}})$ are derived from $c_{\text{joint}}$ and $c_{\text{clip}}$ using linear projection layers, respectively. By integrating high-level relational reasoning ($c_{\text{joint}}$), semantic identity guidance ($c_{\text{clip}}$), and low-level appearance details ($c_{\text{vae}}$) in this structured manner, BindWeave effectively steers the diffusion process to generate videos that are not only visually faithful to the subjects but also logically and semantically aligned with complex user instructions.

### 3.5 TRAINING AND INFERENCE

**Training Setup.** Following the rectified flow formulation described in Section 3.1, our model is trained to predict the ground truth velocity. The overall training objective for BindWeave can be formulated as mean squared error (MSE) between the model output and $v_t$:

$$\mathcal{L}_{\text{mse}} = \|u_\Theta(z_t, t, c_{\text{joint}}, c_{\text{clip}}, c_{\text{vae}}) - v_t\|_2^2. \qquad (13)$$

Our training data is curated from the 5 million publicly available OpenS2V-5M dataset (Yuan et al., 2025). Through a series of filtering strategies, we distill a final, high-quality dataset of approximately 1 million video-text pairs. The training process then follows a two-stage curriculum learning strategy on this data. All training processes are conducted on 512 xPUs, with a global batch size of 512, utilizing a constant learning rate of 5e-6 and the AdamW optimizer. The initial stabilization phase, lasting for approximately 1,000 iterations, utilizes a smaller, core subset selected from the 1 million data for its exceptional quality and representativeness. This initial phase is crucial for adapting the model to the specific demands of the Subject-to-Video (S2V) task, primarily focusing on learning to faithfully preserve a subject's visual identity while aligning it with the corresponding textual motion commands. This establishes a robust foundation for the subsequent large-scale training. Subsequently, the training transitions to a full-scale phase for an additional 5,000 iterations, where the model is exposed to the entirety of the 1 million curated dataset. This second stage allows the model to build upon its stable foundation and learn from a broader range of high-quality examples, significantly enhancing its generative capabilities and generalization performance.

**Inference settings.** During inference, our BindWeave accepts a flexible number of reference images (typically 1-4), while a text prompt steers the generation by describing the desired scene and behaviors. Similar to Phantom (Liu et al., 2025), we use a prompt rephraser during inference to ensure the text accurately describes the provided reference images. Generation is performed over 50 steps using a rectified flow Liu et al. (2022) trajectory, guided by Classifier-Free Guidance (CFG) Ho & Salimans (2022) with a scale of $\omega$. The guided noise estimate at each step $t$ is computed as:

$$\hat{\epsilon}_\theta(x_t, c) = \epsilon_\theta(x_t, \oslash) + \omega\left(\epsilon_\theta(x_t, c) - \epsilon_\theta(x_t, \oslash)\right) \qquad (14)$$

where $\epsilon_\theta(x_t, c)$ is the noise prediction conditioned on the prompt $c$, and $\epsilon_\theta(x_t, \oslash)$ is the unconditional prediction. This estimate is then used by the scheduler to derive $x_{t-1}$.

# 4 EXPERIMENTS

## 4.1 EXPERIMENTAL SETTINGS

**Benchmark and Evaluation Metrics.** To ensure a fair comparison, we adopt the OpenS2V-Eval benchmark (Yuan et al., 2025) and follow its official evaluation protocol, which provides fine-grained assessments of subject consistency and identity fidelity for subject-to-video generation. The benchmark comprises 180 prompts in **seven distinct categories**, covering scenarios from single-subject (face, body, entity) to multi-subject and human–entity interactions. To quantify performance, we report the protocol's automated metrics, with higher scores indicating better results across all metrics. These include **Aesthetics** (christophschuhmann, 2024) for visual appeal, **MotionSmoothness** (Bradski et al., 2000) for temporal smoothness, **MotionAmplitude** (Bradski et al., 2000) for motion magnitude, and **FaceSim** (Yuan et al., 2024a) for identity preservation. We also use three metrics introduced by OpenS2V-Eval (Yuan et al., 2025) that correlate highly with human perception: **NexusScore** (subject consistency), **NaturalScore** (naturalness), and **GmeScore** (text–video relevance).

**Implementation details.** BindWeave is fine-tuned from a foundation video generation model based on DiT architecture (Wan et al., 2025). The T2V and I2V pre-training stages are excluded from this evaluation. For the core instruction planning module, we employ Qwen2.5-VL-7B (Bai et al., 2025) as our Multimodal Large Language Model (MLLM). To align the multimodal control signal with the DiT conditioning space, we introduce a lightweight connector that projects the Qwen2.5-VL hidden states. Specifically, the connector features a two-layer MLP with GELU activation. We train our model using the Adam optimizer with a 5e-6 learning rate and a global batch size of 512. To mitigate copy-paste artifacts, we apply data augmentations (e.g., random rotation, scaling) to reference images. During inference, we use 50 denoising steps set the CFG guidance scale $\omega$ to 5.

**Baselines.** We compare BindWeave with the state-of-the-art video customization methods, including open-sourced methods (Phantom (Liu et al., 2025), VACE (Jiang et al., 2025), SkyReels-A2 (Fei et al., 2025), MAGREF (Deng et al., 2025)) and commercial products (Kling-1.6 (Kwai, 2024), Vidu-2.0 (Bao et al., 2024), Pika (Lab, 2024), Hailuo (Team, 2024)).

## 4.2 QUANTITATIVE RESULTS

We conduct a comprehensive comparison on the OpenS2V-Eval benchmark (Yuan et al., 2025), as shown in Table 1, providing a broad and rigorous evaluation across diverse scenarios. Following the benchmark's protocol, each method generates 180 videos for evaluation to ensure statistical reliability and coverage of all categories. We report eight automatic metrics as described in Section 4.1 to ensure comprehensive assessment, thereby capturing visual quality, temporal behavior, and semantic alignment in a unified manner. As shown in Table 1, our BindWeave achieves a new state of the art on the overall Total Score, with notably stronger NexusScore that highlights its advantage on subject consistency. Notably, NexusScore (Yuan et al., 2025) is designed to address the limitations of prior global-frame CLIP (Radford et al., 2021a) or DINO (Oquab et al., 2023) comparisons and provide a semantically grounded, noise-resilient assessment that better reflects perceptual identity fidelity. It achieves this via a detect-then-compare strategy that first localizes the true target, crops the relevant regions to suppress background interference, and then computes similarity within a retrieval-based multimodal feature space with confidence and text–image gating, finally aggregating scores over the verified crops for a reliable summary. Importantly, BindWeave also maintains strong competitiveness on other metrics, including FaceSim, Aesthetics, GmeScore, motion-related measures such as MotionSmoothness and MotionAmplitude, and NaturalScore, which respectively reflect its strengths in identity preservation, visual appeal, text–video alignment, temporal coherence and motion magnitude, and overall naturalness across a wide range of prompts and categories.

## 4.3 QUALITATIVE RESULTS

To clearly demonstrate the effectiveness of our method, we present some typical subject-to-video scenarios in Figure 4 and Figure 5, including single-body-to-video, human-entity-to-video, single-object-to-video, and multi-entity-to-video. As shown in the left panel of Figure 4, commercial models such as Vidu, Pika, Kling, and Hailuo produce visually appealing videos but struggle with subject consistency. Among open-source methods, SkyReel-A2 is comparatively competitive on

Table 1: Quantitative comparison among different methods for subject-to-video task. Total score is the normalized weighted sum of other scores. "↑" higher is better.

| Method | Total Score↑ | Aesthetics↑ | MotionSmoothness↑ | MotionAmplitude↑ | FaceSim↑ | GmeScore↑ | NexusScore↑ | NaturalScore↑ |
|---|---|---|---|---|---|---|---|---|
| VACE-14B Jiang et al. (2025) | 57.55% | 47.21% | 94.97% | 15.02% | **55.09**% | 67.27% | 44.08% | 67.04% |
| Phantom-14B Liu et al. (2025) | 56.77% | 46.39% | 96.31% | 33.42% | 51.46% | 70.65% | 37.43% | 69.35% |
| Kling1.6(20250503) Kwai (2024) | 56.23% | 44.59% | 86.93% | **41.60**% | 40.10% | 66.20% | 45.89% | **74.59**% |
| Phantom-1.3B Liu et al. (2025) | 54.89% | 46.67% | 93.30% | 14.29% | 48.56% | 69.43% | 42.48% | 62.50% |
| MAGREF-480P Deng et al. (2025) | 52.51% | 45.02% | 93.17% | 21.81% | 30.83% | 70.47% | 43.04% | 66.90% |
| SkyReels-A2-P14B Fei et al. (2025) | 52.25% | 39.41% | 87.93% | 25.60% | 45.95% | 64.54% | 43.75% | 60.32% |
| Vidu2.0(20250503) Bao et al. (2024) | 51.95% | 41.48% | 90.45% | 13.52% | 35.11% | 67.57% | 43.37% | 65.88% |
| Pika2.1(20250503) Lab (2024) | 51.88% | 46.88% | 87.06% | 24.71% | 30.38% | 69.19% | 45.40% | 63.32% |
| VACE-1.3B Jiang et al. (2025) | 49.89% | **48.24**% | **97.20**% | 18.83% | 20.57% | 71.26% | 37.91% | 65.46% |
| VACE-P1.3B Jiang et al. (2025) | 48.98% | 47.34% | 96.80% | 12.03% | 16.59% | **71.38**% | 40.19% | 64.31% |
| **Ours** | **57.61**% | 45.55% | 95.90% | 13.91% | 53.71% | 67.79% | **46.84**% | 66.85% |

subject consistency, yet its overall visual aesthetics lag behind our BindWeave. VACE and Phantom similarly exhibit weak subject consistency. In the right panel of Figure 4, our approach achieves markedly better subject consistency, text alignment, and visual quality. As shown in the left panel of Figure 5, in single-object-to-video scenarios, commercial models such as Vidu and Pika still exhibit pronounced violations of physical and semantic plausibility—what we summarize as "common-sense violations" (e.g., a human walking with severely twisted legs). Kling achieves strong visual aesthetics but maintains poor subject consistency. SkyReels-A2 shows severe distortions and similarly weak subject consistency, and Phantom also struggles to preserve subject consistency. Among the baselines, VACE better maintains subject consistency but suffers from limited motion coherence and naturalness. In contrast, our BindWeave delivers strong subject consistency together with natural and coherent motion. Notably, under multi-object and complex-instruction settings as shown in the right panel of Figure 5, methods like Vidu and Pika often miss key cues (e.g., "hot oil"), Kling shows severe physical implausibility (e.g., fries leaking directly out of the basket), and MAGREF fails to preserve subject consistency; other baselines also omit crucial prompt details. In contrast, our results deliver fine-grained detail while maintaining strong subject consistency. We attribute this to BindWeave's explicit cross-modal integration of the reference image and textual prompt via an MLLM, which jointly parses entities, attributes, and inter-object relations. As a result, BindWeave preserves subtle yet crucial details (e.g., "hot oil") and constructs a unified, temporally consistent scene plan to guide coherent generation. This deep cross-modal integration reliably enforces key prompt elements and embeds basic physical commonsense for multi-entity interactions, thereby reducing implausible outcomes. More visualizations can be found in Appendix Section A.5 and our supplementary materials, including additional qualitative examples and comparisons.

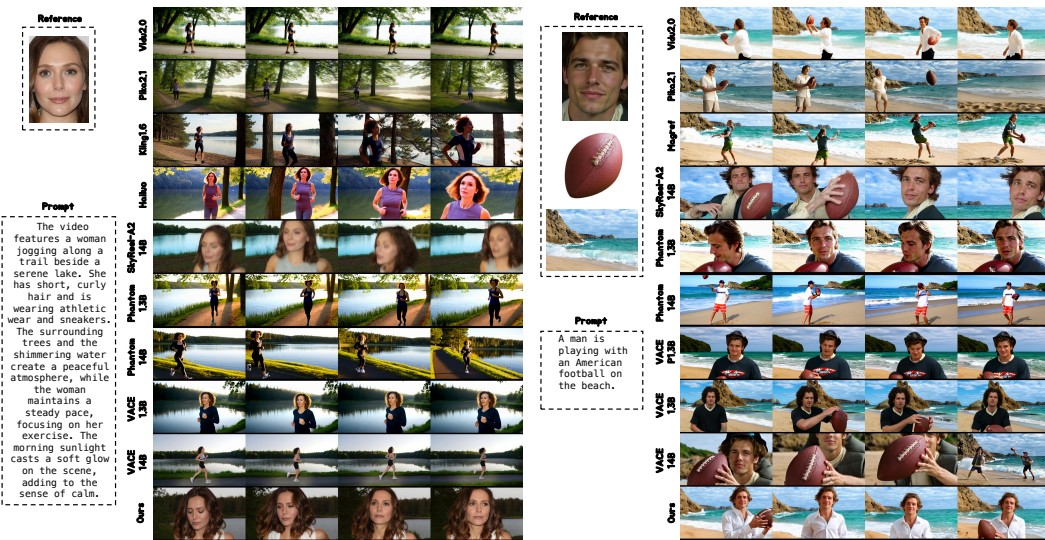

Figure 4: Qualitative comparison on subject-to-video task, with four uniformly sampled frames shown in each case. Compared to other competing methods, our approach is superior in subject fidelity, naturalness, and semantic consistency with the caption.

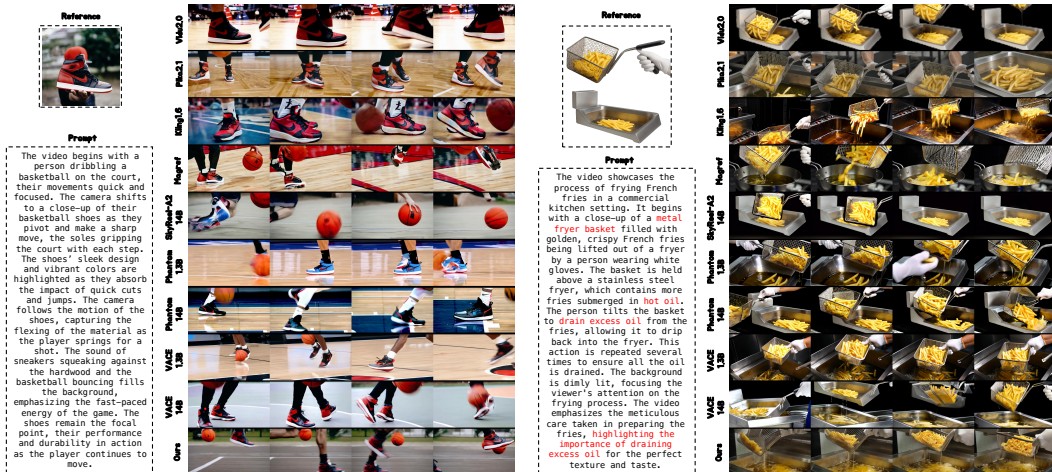

Figure 5: Qualitative comparison on subject-to-video task, with four uniformly sampled frames shown in each case. Compared with other methods, our approach better avoids implausible phenomena and produces more natural videos while maintaining strong subject consistency.

Table 2: Quantitative ablation results comparing T5-only and T5+Qwen2.5-VL conditioning.

| Method | Total Score↑ | Aesthetics↑ | MotionSmoothness↑ | MotionAmplitude↑ | FaceSim↑ | GmeScore↑ | NexusScore↑ | NaturalScore↑ |
|---|---|---|---|---|---|---|---|---|
| T5-only | 55.16% | 42.80% | 95.39% | 7.48% | 53.02% | 62.26% | 45.79% | 63.38% |
| T5+Qwen2.5-VL | **57.61%** | **45.55%** | **95.90%** | **13.91%** | **53.71%** | **67.79%** | **46.84%** | **66.85%** |

## 4.4 ABLATION STUDY

We ablate our control-conditioning that concatenates MLLM- and T5-derived signals to guide a DiT during generation. We compare a T5-only baseline with our T5+Qwen2.5-VL variant. The MLLM-only setup (Qwen2.5-VL + DiT) is omitted from this quantitative analysis because, as discussed in Section A.3, it proved to be unstable during training and failed to converge within our training budget. As shown in Table 2, T5+Qwen2.5-VL consistently outperforms T5-only across aesthetics, motion, naturalness, and text relevance. Qualitative comparisons in Figure 6 further corroborate these findings: when reference images exhibit scale mismatch, the T5-only baseline tends to produce unrealistic subject sizes (e.g., dog–bowl), and under complex instructions it often misparses action–object relations, whereas T5+Qwen2.5-VL remains well grounded and executes the intended interactions. We attribute these gains to complementary conditioning, the MLLM provides multimodal, identity- and relation-aware cues that disambiguate subjects and improve temporal coherence, while T5 offers precise linguistic grounding that stabilizes optimization. Their concatenation yields a richer and more reliable control signal for DiT. More cases are provided in Section A.4.

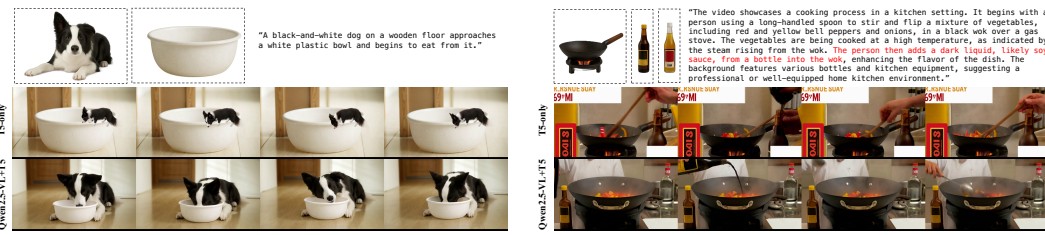

Figure 6: Qualitative comparison of MLLM+T5 vs. T5-only. MLLM+T5 shows superior scale grounding, reliable action–object execution, and stronger temporal/textual coherence.

## 4.5 USER STUDY

We conducted a user study to evaluate our method, employing mean opinion scores (MOS) across four key criteria: subject consistency, prompt following, video quality, and motion quality. These dimensions provide a comprehensive assessment of the generated videos. We recruited 20 participants to perform anonymized comparisons across different methods. All generated samples are

| Method | Subject Consistency ↑ | Prompt Following ↑ | Motion Quality ↑ | Video Quality ↑ | Total Score Average ↑ |
|---|---|---|---|---|---|
| SkyReels | 3.47 | 3.52 | 3.60 | 3.23 | 3.46 |
| Vidu | 3.40 | 3.55 | 3.63 | 3.50 | 3.52 |
| Magref | 3.25 | 3.65 | 3.65 | 3.58 | 3.53 |
| Phantom | 3.56 | **3.72** | 3.84 | 3.58 | 3.68 |
| Kling | 3.62 | 3.58 | **3.90** | **3.75** | 3.71 |
| VACE | 3.88 | 3.63 | 3.80 | 3.62 | 3.73 |
| **BindWeave (Ours)** | **3.94** | 3.66 | 3.75 | 3.70 | **3.76** |

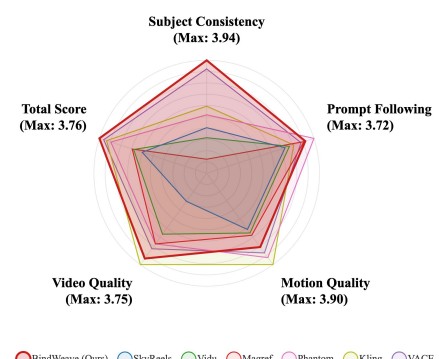

Table 3: User study results comparing different methods. "Total Score" means the average score.

Figure 7: Visualization of user study scores across the evaluation criteria.

anonymized and randomly assigned to users, with each item scored on a 1–5 scale. The results are summarized in Table 3, and for clearer presentation we also visualize the data in Figure 7. The evaluation indicates that BindWeave achieves the best performance in subject consistency while maintaining leading results across all metrics.

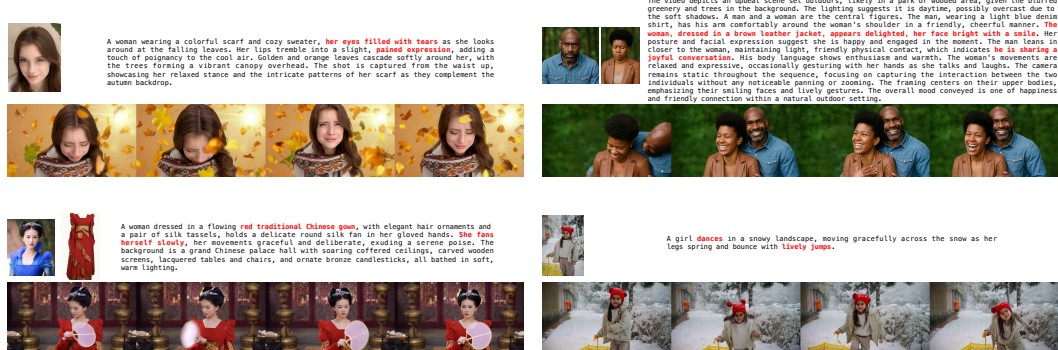

Figure 8: Evaluating copy–paste artifacts under conflict–coherence scenarios

## 4.6 COPY-PASTE PROBLEM

We evaluate whether conditioning on reference images induces pixel-level "copy-paste" artifacts by constructing conflict-coherence scenarios that deliberately mismatch the reference and the prompted outcome in Figure 8. Beyond contrasting facial expressions (e.g., smiling reference → painful, tearful video; painful reference → joyful smiling video), we also test outfit changes and diverse pose settings to more comprehensively validate our approach. Across these cases, the model follows the instructions rather than pasting pixels; it adapts facial musculature, attire, and pose to match the prompt while preserving identity, maintains smooth temporal transitions without stuck frames, and produces coherent motion instead of static overlays.

## 5 CONCLUSION

In this paper, we introduce BindWeave, a novel subject-consistent video generation framework that delivers consistent, text-aligned, and visually compelling videos across single- and multi-entity settings through explicit cross-modal integration. By using an MLLM to deeply integrate information from reference images and textual prompts to facilitate joint learning, BindWeave effectively models entity identities, attributes, and relations, thereby achieving fine-grained grounding and strong subject preservation. The empirical results demonstrate that BindWeave has fully learned cross-modal fusion knowledge, enabling the generation of high-fidelity, subject-consistent videos. Moreover, on the OpenS2V benchmark, BindWeave achieves state-of-the-art performance, outperforming existing open-source methods and commercial models, clearly showcasing its strength. Overall, BindWeave offers a new perspective for the S2V task and points toward future advances in consistency, realism, and controllability.

ACKNOWLEDGMENTS

This work was partially supported by the National Key R&D Program of China (Grant No. 2024YFB3909902) and Youth Innovation Promotion Association of CAS.

ETHICS STATEMENT

This work studies subject-to-video generation and related evaluation. All images appearing in this paper are either generated by our models or sourced from publicly available datasets under their respective licenses and are used solely to demonstrate the technical capabilities of our research. All qualitative (visualized) results are provided solely for academic comparison and research discussion and are not intended for commercial use.

REPRODUCIBILITY STATEMENT

In Section 3.2, we provide a detailed description of our network architecture and the interactions among the variables. In Section 3.5, we disclose the detailed parameters for training and inference, as well as the datasets used. In Section 4.1, we further present our model configurations and the parameters for training and inference, including the benchmarks and metrics used for performance evaluation. Through these efforts, we have made every possible attempt to ensure the reproducibility of our method. Furthermore, we will open-source our code and models to facilitate reproducibility.

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

## A    APPENDIX

### A.1    APPENDIX OVERVIEW

This appendix comprises two sections:

- **Section A.2: LLM Usage Disclosure.**  We disclose that we used large language models (LLMs) solely for minor grammar checking after drafting the manuscript, with no contribution to research ideation, methods, experiments, or results.
- **Section A.3: Justification for Model Design.**  In this section, we discuss our architectural choices and provide a clear, empirical justification for our final model design.
  **Section A.4: Visual comparisons in complex multi-subject scenarios between T5-only and MLLM+T5.**  We present extended qualitative comparisons under complex multi-subject settings.
- **Section A.5: Additional Subject-to-Video Quantitative Results.**  We present extended quantitative results for the subject-to-video setting.

### A.2    LLM USAGE DISCLOSURE

We use large language models (LLMs) solely for minor grammar checking after drafting the manuscript. LLMs did not contribute to research ideation, problem formulation, method design, theoretical development, experiments, implementation, result analysis, or the creation of figures/tables. No code, data, or experimental results were generated by LLMs. All LLM-suggested edits were manually reviewed and verified by us. We did not provide any LLM with non-public or sensitive information.

### A.3    JUSTIFICATION FOR MODEL DESIGN

During the development phase, we conducted a series of pilot experiments to explore various architectural designs for integrating the Multi-modal Large Language Model (MLLM) and the Diffusion Transformer (DiT). These explorations served as an informal ablation study and directly informed our final design choice. Specifically, we investigated several alternatives to our proposed **MLLM + MLP + T5 + DiT** architecture, including:

1. **MLLM + MLP + DiT:** Using a simple Multi-Layer Perceptron (MLP) to project MLLM features into the DiT.
2. **MLLM + Q-Former + DiT:** Employing a Q-Former structure to refine the MLLM outputs before feeding them to the DiT.

Our key finding was that architectures relying solely on the MLLM for conditioning (i.e., those without the T5 text encoder) were **unstable during training and failed to converge within our available training budget.**  These models struggled to effectively translate the nuanced visual identity information from the MLLM into the precise conditional inputs required by the DiT for high-fidelity video generation. In contrast, the inclusion of T5 provided a robust and stable text-conditioning backbone, which, when combined with the identity features from the MLLM, yielded the consistent and high-quality results presented in our paper. To provide concrete evidence of the convergence issues, we have included a training loss curve for the **MLLM + MLP + DiT** architecture (Figure 9). This curve clearly illustrates the training instability, showing that the model's loss oscillates significantly and fails to reach convergence.

### A.4    COMPARATIVE RESULTS OF T5-ONLY VERSUS MLLM+T5 (BINDWEAVE) UNDER COMPLEX SCENARIOS

Figure 10 presents extensive challenging cases, including five-image reference setups and complex multi-subject interactions, with side-by-side comparisons between T5-only and MLLM+T5. Across these multi-reference cases, the T5-only model frequently exhibits distortions and temporal jitter, leading to unstable videos. In contrast, MLLM+T5 reliably reasons about subject placement and overall layout, yielding coherent spatial arrangements, stable motion, and higher visual quality. In

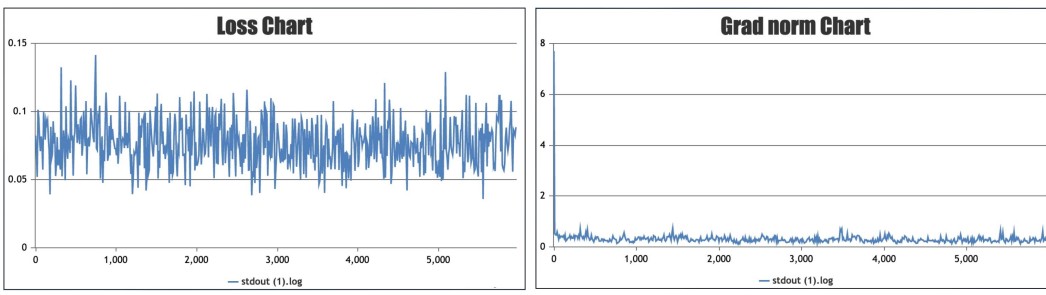

Figure 9: Training loss for the **MLLM + MLP + DiT** architecture, showing significant oscillation and failure to converge.

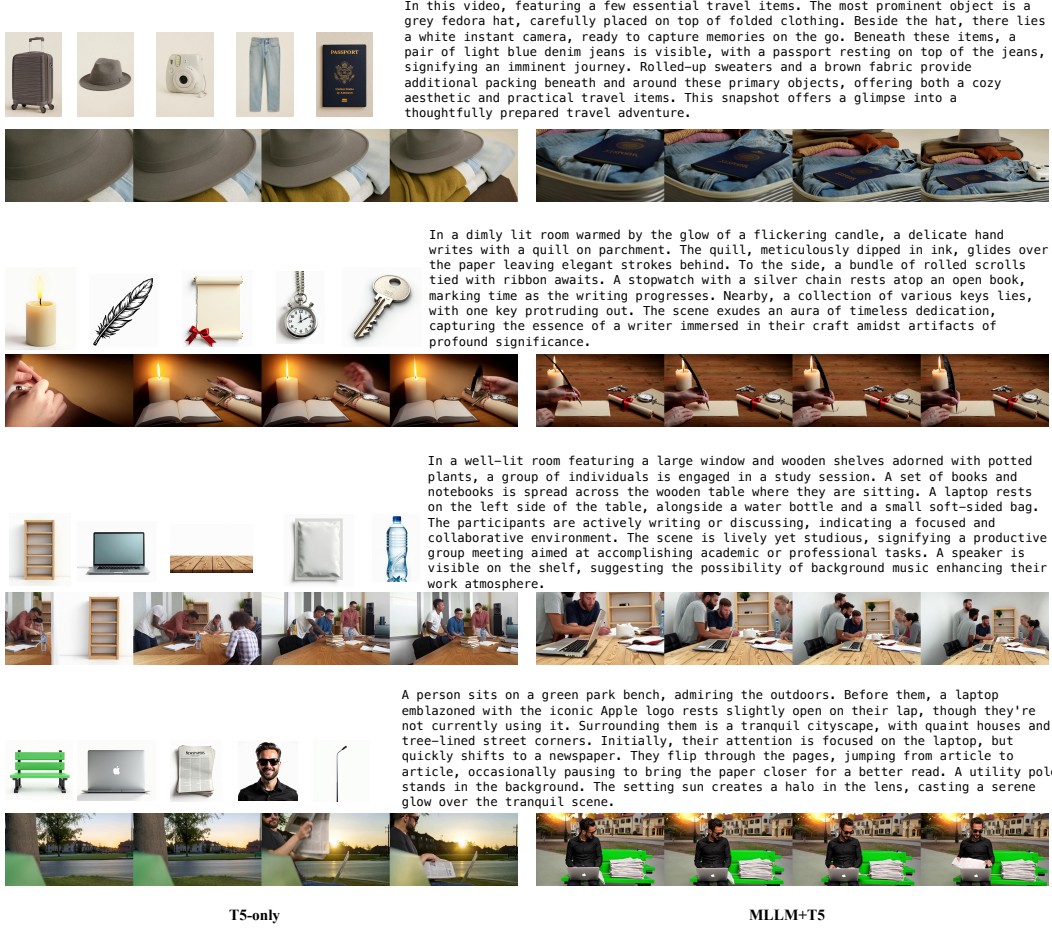

Figure 10: Qualitative comparison of T5-only vs. MLLM+T5.

essence, T5 provides dense captions that help structure prompt interpretation and offer a stable descriptive prior. However, relying on T5 alone is insufficient for compositional spatial/temporal reasoning and multi-subject role disambiguation: T5-only often misses cross-entity relations, produces inconsistent event ordering, and may conflate subject roles—leading to layout instability and drift in videos. The MLLM supplies compositional reasoning and cross-entity constraint satisfaction. With MLLM+T5, the model correctly resolves ordered cross-entity relations and better maintains relative positions and action dependencies.

## A.5 MORE SUBJECT-TO-VIDEO QUANTITATIVE RESULTS

We present a set of comparative cases, as shown in Figure 11, in which the prompt specifies "a man" whereas the reference image depicts a baby. Under this conflict, many methods (Vidu 2.0 Bao et al. (2024), Pika 2.1 Lab (2024), Phantom Liu et al. (2025), VACE Jiang et al. (2025).) generate a man as the subject rather than a baby. Other methods, such as Kling 1.6 Kwai (2024), Hailuo and SkyReel-A2, retain some infant characteristics but still exhibit poor overall consistency with the reference image. In contrast, our method is not perturbed by the prompt and faithfully preserves the baby's appearance from the reference image.

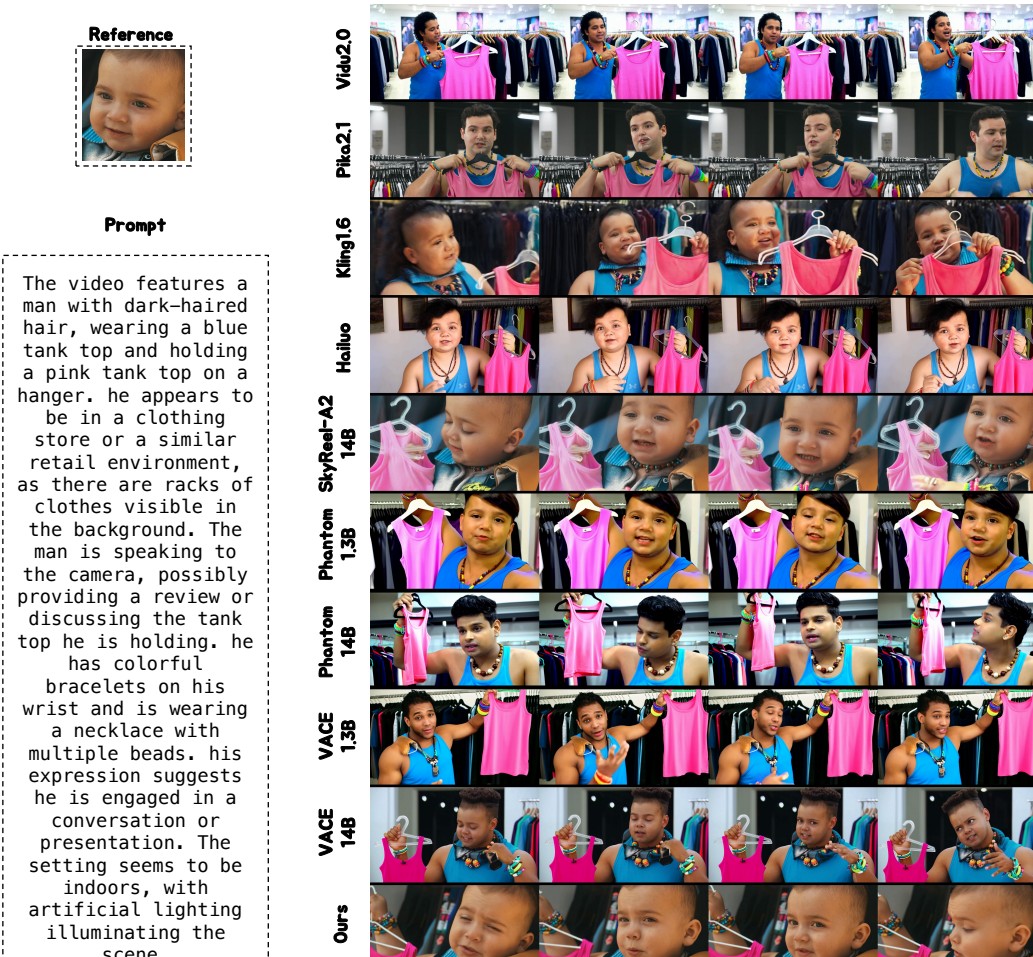

Figure 11: Comparisons under a prompt–reference ambiguity (prompt: "a man"; reference: baby). Most baselines follow the prompt and generate an adult male, ignoring the features of the reference image or retaining only partial infant traits, whereas our method faithfully preserves the reference subject's appearance.

As shown in Figure 12, under a relatively simple prompt, most baseline methods exhibit pronounced copy-paste issues: the subject remains static across frames. In particular, Phantom-1.3B Liu et al. (2025) and VACE-14B Jiang et al. (2025) essentially copy-paste the reference cat directly into the video. In contrast, our method avoids the copy–paste issue while preserving subject consistency, yielding natural and temporally coherent motion.

As shown in Figure 13, given only a single face reference image, BindWeave generates high-fidelity, subject-consistent videos. It preserves fine-grained identity cues, including facial structure, skin tone, hairstyle, while handling changes in pose, expression, and moderate viewpoint or illumination.

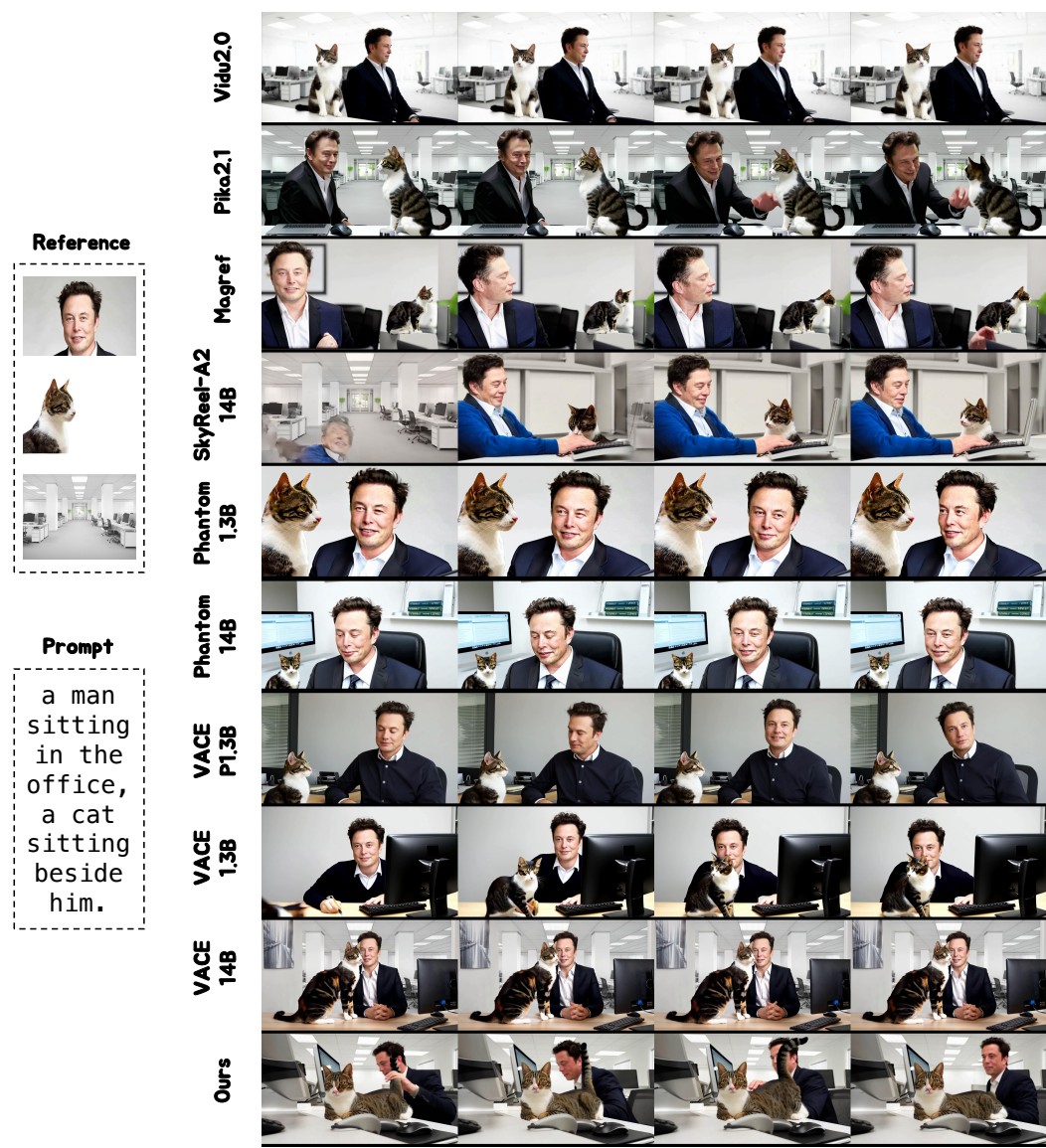

Figure 12: Comparisons in the subject-to-video setting illustrating the copy–paste issue under simple prompts. Many baselines directly copy the reference image into the video, causing the subject to remain static across frames, whereas our method preserves the subject's temporal dynamics and natural motion.

The results avoid copy-paste artifacts and identity drift, delivering smooth, temporally coherent motion and maintaining alignment with the text prompt without overriding the reference appearance.

As shown in Figure 14, when provided with multiple reference subjects, BindWeave maintains consistent identity for each subject across frames and preserves fine-grained appearance details as well as their relative spatial layout. The method produces natural, coordinated interactions between subjects, keeps compositions visually pleasing and realistic, and avoids identity swapping or blending.

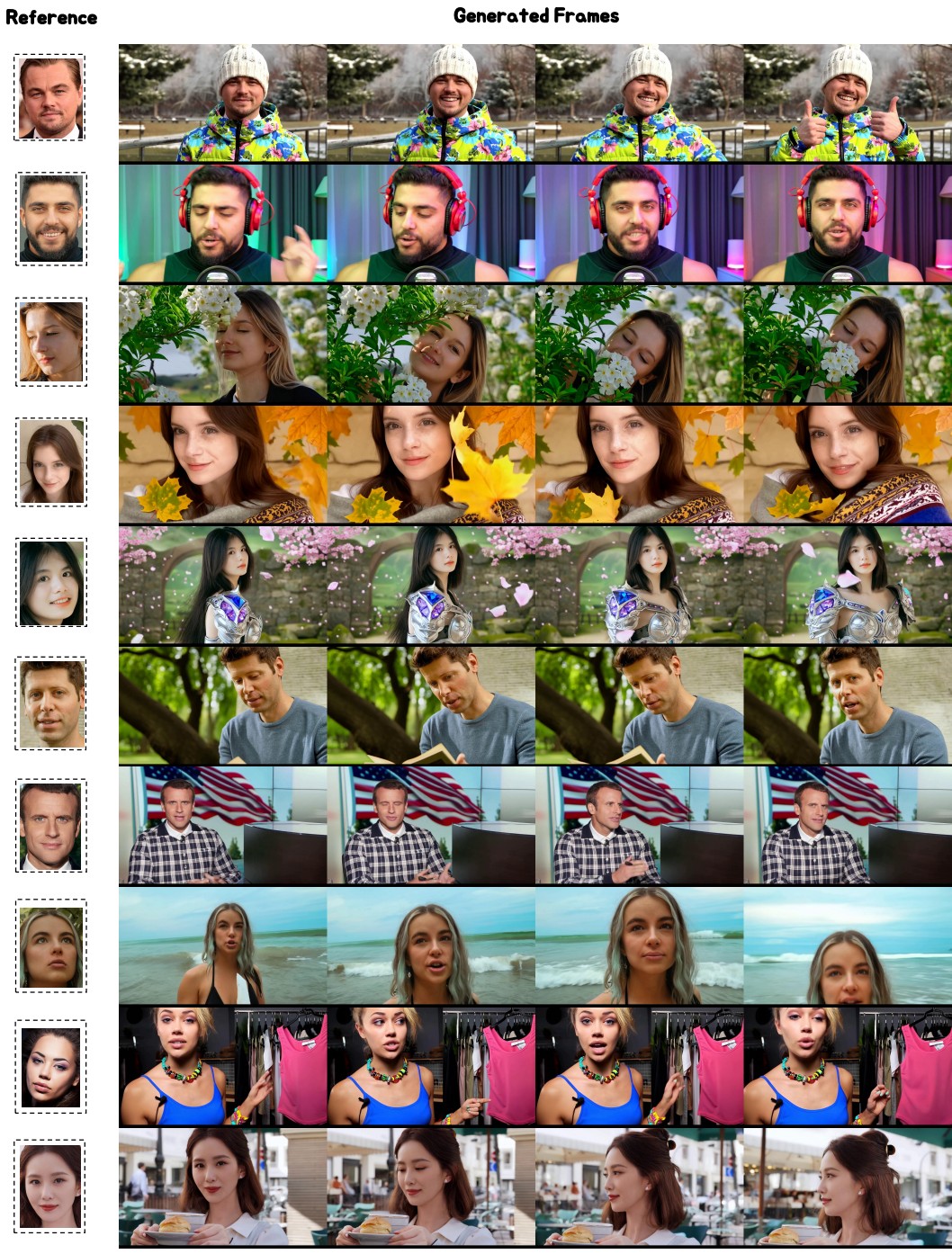

Figure 13: More generated results of BindWeave, demonstrating high fidelity and strong subject consistency.

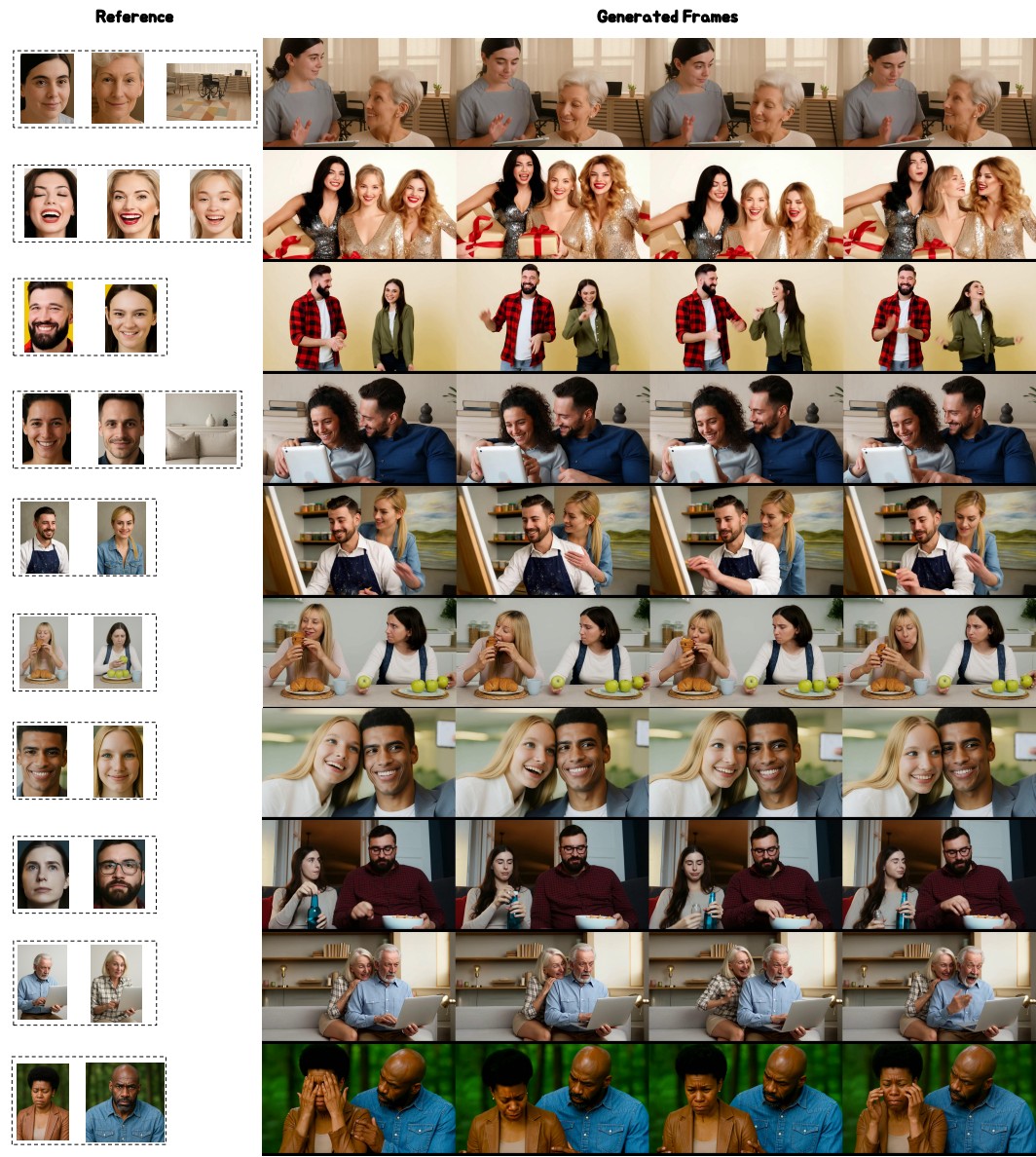

Figure 14: More generated results of BindWeave, demonstrating high fidelity and strong subject consistency across multi-references.

