# OpenReview forum: "BindWeave: Subject-Consistent Video Generation via Cross-Modal Integration"
_ICLR.cc/2026/Conference — ICLR 2026 Poster_

### Official Review · Reviewer_NFCf · 2025-10-19

**Soundness:** 3
**Presentation:** 4
**Contribution:** 2
**Rating:** 4
**Confidence:** 5

**Summary:**

The paper introduces BindWeave, a subject-consistent video generation framework that leverages deep cross-modal integration between text and image inputs. The authors claim that the existing models typically fail to handle complex multi-subject interactions. To overcome this, BindWeave integrates a MLLM with a DiT. Specifically, the MLLM is used to encode spatio-temporal relationships into subject-aware hidden states, which then condition the diffusion process for video synthesis. The experiments are conducted on the OpenS2V benchmark and show BindWeave outperforming state-of-the-art open-source and commercial baselines.

**Strengths:**

- This paper targets at subject-to-video generation which is an interesting and practical problem relevant to lots of real-world application, such as video personalization or content creation.
- The proposed adaptive multi-reference conditioning strategy sounds well-motivated to me as it allows the model to flexibly integrate multiple reference images without disrupting temporal coherence.
- The model is trained on the open-sourced OpenS2V-5M dataset, and the paper provides thorough implementation details which could enhance the reproducibility of the proposed framework.
- The experiments include a comprehensive comparisons with 9 open-sourced and proprietary baseline models. The quantitive results on the OpenS2V-Eval benchmark look promising and show consistent improvements from some existing models.
- The writing quality is high and easy to understand. The figures are also well-plotted which can assist the readers to quickly understand the proposed framework.

**Weaknesses:**

- My major concern is the limited technical contribution. The framework jointly using "frozen MLLM & trainable connector" + "VAE tokens" + "noisy latents" to condition the diffusion process is a very common design in recent editing models, such as Step1X-Edit [1] and Qwen-Image [2]. The framework figures (i.e., Figure 2 of BindWeave paper and Figure 4 of Step1X-Edit paper) are nearly the same. It significantly limits the technical contribution/novelty of the paper.
- While one of the main claims of the paper is the usage of the powerful MLLM for cross-modal reasoning, most of the provided samples (e.g., "[man] holds a [ball]", "[sauces] placed nearby a [wok]"), however, do not require complex reasoning and can be simply achieved by existing models. While Figure 6 shows the comparison of the samples using T5-only and MLLM+T5, the results do not look reasonable to me as I never saw a model combining [dog] and [bowl] in this way even without MLLM reasoning capability.
- Following the above concern, the authors should provide samples to demonstrate the effectiveness of MLLM in handling complex multi-subject interactions that require spatial-temporal reasoning. For example, a prompt like "[basketball court] and four [basketball players]", where two teammates wear same uniforms and one is attacking while the others are defending. In this sample, the model should correctly place the two teammates on the same side and assign proper actions to each player to achieve spatial-temporal coherence in the output video.
- Similar to the existing editing frameworks, one major limitation of using VAE tokens as condition is the presence of copy-paste artifacts. In the provided samples, I also noticed heavy pixel-level alignment between the conditional images and the output videos. To further verify whether BlindWeave suffers from this issue, the authors should include samples with conflicting "image-image" or "image-text" coherence. For example, "a [garment image] and a [person image] with different poses" or "make a [crying boy image] [smile (text)]"
- Following the concern above, the model seems to achieve poor trade-off between "pose variation" and "fidelity". For example, Figure 7 shows high fidelity but low pose variation (i.e., heavy pixel-level alignment), while Figure 8 shows huge pose variation but low fidelity. Based on these results, I don't think BlindWeave can solve these copy-paste artifacts well.
- The scope of ablation study is very limited. There is only one ablation (T5-only vs. T5+MLLM), while it is also interesting to check the effect of isolating CLIP, VAE, or individual conditional streams.
- The authors do not provide supplementary material with video samples, which makes the reviewers unable to justify the quality of the output videos. It is impossible to check the temporal smoothness simply based on the provided video frames.

[1] "Step1X-Edit: A Practical Framework for General Image Editing"

[2] "Qwen-Image Technical Report"

**Questions:**

- Could the author provide more samples that require complex spatial-temporal reasoning to verify the effectiveness of the MLLM model?
- Could the author provide more samples with conflicting conditions which can verify whether BlindWeave suffers from copy-paste artifacts?
- Could the author provide more detailed ablation study to verify the effectiveness of each module in the framework?
- The model performance looks inconsistent in Figures 7 and 8. Could the authors explain this phenomena? Does the model include a mechanism supporting the adjustment of the trade-off between "fidelity" and "pose variation"?
- [Minor] The citation format throughout the paper is wrong. For example, should be "Phantom (Liu et al., 2025)" instead of "Phantom Liu et al. (2025)". You can modify it by changing "cite"/"citet" to "citep"
- [Minor] In line 210: should be "we introduce" instead of "we introduces"

---

> ### Author Response · Authors · 2025-11-21
> **Response to Reviewer NFCf (Part 1/2)**
>
> We sincerely appreciate your rigorous review and valuable insights. Your observations have been instrumental in refining this work. Below, we respond to each of your comments with our detailed clarifications and updates.
>
> ### **Response 1: About technical contribution**
> Thank you for the detailed comparison of BindWeave with recent image editing frameworks. We appreciate the context and agree that "frozen MLLM + trainable connector + VAE tokens + noisy latents" has become a common pattern in image generation/editing. However, our work targets a distinct problem setting and introduces a non-trivial integration that, to the best of our knowledge, is novel in subject-consistent video generation.
> - Problem focus and scope:
>   - Step1X-Edit and Qwen-Image are primarily designed for image generation or image editing. In contrast, our method specifically addresses subject-consistent video generation with multi-reference, multi-subject conditioning across time, where identity stability and relational coherence are critical.
>   - Temporal consistency and cross-frame identity anchoring introduce challenges not present in single-image pipelines, requiring a different conditioning strategy and fusion design to avoid drift.
> - Key novelty in BindWeave:
>   - Unified conditioning with MLLM + T5: In our pipeline, the frozen MLLM’s hidden states are first projected by an MLP and then concatenated with T5 text encoder features, forming a stable and comprehensive conditioning anchor for the DiT backbone. This design provides complementary strengths—MLLM for relational and identity-aware cues from multiple references; T5 for stable, precise, and controllable text conditioning—leading to robust convergence and reduced drift.
>   - Subject-grounded multi-reference fusion: We explicitly handle multiple references via concatenation and learned fusion, using CLIP features for global semantic anchoring and VAE tokens for low-level appearance priors. This tri-modal, subject-grounded conditioning is tailored for video identity consistency rather than image editing.
>   - Practical training and convergence insights: In our revised manuscript (Section A.3), we report that MLLM-only conditioning proved unstable and did not converge in our subject-consistent video generation setting. These empirical findings motivated our final architecture and illustrate why naively transferring image-editing frameworks is insufficient for this task.
>
> In summary, our contributions lie in targeting a different task regime (subject-consistent video generation), proposing a complementary MLLM+T5 fusion tailored for temporal identity stability, and demonstrating that direct transplantation of image-editing architectures is inadequate for this problem.
>
> ### **Response 2: Comparison of T5-only vs. MLLM+T5 in More Complex Scenarios**
> Thank you for the thoughtful feedback.
> - Empirical observation on T5-only: After multiple independent runs, our T5-only baseline consistently produced results for the [dog] and [bowl] case similar to those shown in the paper. Our claim is not that simple relations are impossible for prior models, but that the MLLM features provide more reliable relational grounding across challenging cases and reduce failure rates under subject-consistent video constraints.
> - Additional complex reasoning cases: Following your suggestion, we have added more challenging examples in the supplementary material, including complex multi-subject interactions and various interactions among multiple human and objects. These better illustrate the advantage of MLLM+T5 over T5-only beyond the simpler cases. As noted in our Response 4 to Reviewer rb5c—where we highlighted typical failures in complex multi-subject scenes with close interactions (e.g., swapping/blending when appearances are similar)—we observe that in the suggested :"[basketball court] and four [basketball players]" scenario, identity preservation may be imperfect because the teammates wear the same uniforms; nevertheless, MLLM-enhanced reasoning still helps reliably place the two teammates on the same side and assign appropriate offensive/defensive actions to each player.  The corresponding video comparisons are available on our anonymous page in Section S7: https://resplendent-fairy-5f1a26.netlify.app/

---

> ### Author Response · Authors · 2025-11-21
> **Response to Reviewer NFCf (Part 2/2)**
>
> ### **Response 3: Copy-paste issue discussion**
> We evaluate whether conditioning on reference images induces pixel-level copy–paste artifacts by constructing conflict–coherence scenarios that deliberately mismatch the references and the prompted outcomes. Beyond contrasting facial expressions (e.g., smiling reference → painful/tearful video; painful reference → joyful/smiling video), we also test outfit changes and diverse pose settings to comprehensively validate our approach. Across these cases, the model follows the instructions rather than pasting pixels: it adapts facial musculature, attire, and pose to match the prompt while preserving identity, maintains smooth temporal transitions without stuck frames, and produces coherent motion instead of static overlays. We include these conflict cases and side-by-side comparisons in the supplementary material to further substantiate the absence of copy–paste artifacts (we also added a copy-paste discussion in Section 4.6 of the revised manuscript). The corresponding video comparisons are available on our anonymous page in Section S5: https://resplendent-fairy-5f1a26.netlify.app/
>
>
> ### **Response 3: Ablation CLIP and VAE**
> Thank you for the suggestion. Our paper’s primary focus is subject-consistent video generation under multi-reference, multi-subject settings. We do not claim novelty in each individual conditioning stream (e.g., CLIP or VAE usage); instead, our contribution lies in the unified conditioning design and its efficacy for temporal identity stability. Accordingly, our main ablation in the paper centers on the core question of reasoning and controllability—comparing T5-only versus MLLM+T5—which directly tests the value of introducing MLLM features for cross-modal relational grounding while maintaining stable text control. In the revised manuscript (Section A.3), we expand the discussion of MLLM–DiT fusion strategies, reporting variants such as MLLM+MLP+DiT, MLLM+Q-Former+DiT, and T5-only baselines; our key finding is that architectures relying solely on the MLLM for conditioning (i.e., without the T5 text encoder) were unstable during training and failed to converge within our available training budget.
>
> ### **Response 4: Video samples**
> We acknowledge the need to assess temporal smoothness beyond static frames. In the revised submission, we provide extensive video samples across diverse scenarios in the supplementary material, enabling reviewers to evaluate temporal coherence, identity consistency, and motion quality. For your convenient review, we also provide an anonymous page: https://resplendent-fairy-5f1a26.netlify.app/
>
> ### **Response to Questions:**
> - We appreciate the question. We would like to clarify that Figures 7 and 8 do not exhibit inconsistent performance. The apparent differences arise from showcasing distinct prompt scenarios and reference configurations rather than contradictory behavior of the model. Our model does not include an explicit mechanism to adjust the fidelity–pose variation trade-off. The observed pose variation primarily follows the prompt specification (e.g., action verbs, spatial cues, camera/viewpoint hints).
> - To provide a more comprehensive view across diverse cases, we have included a large number of video examples on our anonymous page https://resplendent-fairy-5f1a26.netlify.app/. These cover varied subjects, poses, and scales, enabling readers to assess consistency under broader conditions.
>
> - Thank you for the suggestions—we will correct the citation style and fix the typo in the revised manuscript.

---

> ### Author Response · Authors · 2025-11-26
>
> Dear Reviewer NFCf,
>
> We hope this message finds you well. As the discussion period is nearing its end, we wanted to ask if we have addressed all your concerns satisfactorily. If there are any additional points or feedback you would like us to consider, please let us know. Your insights are invaluable to us, and we are eager to resolve any remaining issues to further improve our work. Thank you for your time and effort in reviewing our paper.
>
> Best regards,
>
> Authors of BindWeave

---

### Official Review · Reviewer_TvUz · 2025-11-01

**Soundness:** 3
**Presentation:** 2
**Contribution:** 3
**Rating:** 6
**Confidence:** 3

**Summary:**

This paper targets subject consistent text guided video generation. The key idea is to replace the common separate then fuse conditioning paradigm with cross modal integretion using a MLLM that parses the interleaved prompt with reference images sequence into subject-aware hidden states. These are projected and used alongside T5 text embeddings, CLIP identity features, and VAE low-level features to collectively condition a DiT-based video generator. They showed the competitive results on OpenS2V.

**Strengths:**

1. Deep cross modal binding, the MLLM parses text and image jointly, producing subject aware hidden states that encode roles, attributes, interactions, addressing identity confusion and instructed shallow fusion.
2. Strong ablations and qualitative evidence. T5 only and T5 with MLLM quantitative and qualitative eval shows better scale grounding, action object execution, and temporal coherence.

**Weaknesses:**

major:
1. Using image condition to Qwen2.5 / VAE / CLIP looks redundant, Qwen2.5 and VAE makes sense and also shown a part in your ablation, but why we need a additional CLIP image encoder to encode the condition images.
2. The approach heavily depends on a relatively heavy Qwen2.5VL. The paper doesn’t quantify the extra latency and VRAM and how it scales with K references or video length, more detail would be appreciated.
3. involving the MLLM will bring benefits and also the original issue inherit from the MLLM, the planning itself may fail due to lots of reason like hallucination, how these case will affect the subsequence video generation?

minor:
1. line 102, opens2v --> OpenS2V
2. line 351/359 BinWeave  --> BindWeave, the method name is not consistent in this paper.

**Questions:**

see the major weaknesses

---

> ### Author Response · Authors · 2025-11-21
> **Response to Reviewer TvUz**
>
> Thank you for your thorough and thoughtful assessment. Your comments are highly constructive and have guided several meaningful revisions. We have carefully considered each remark and provide point-by-point responses below.
>
> ### **Response 1: Necessity of the CLIP image encoder**
> Thank you for the thoughtful question. While Qwen2.5-VL (MLLM) and VAE are core to our conditioning pipeline, the CLIP image encoder serves a complementary role that addresses identity stability and semantic grounding:
> - CLIP provides subject-grounded, global semantic anchors from the reference images, which helps reduce identity drift during generation, especially across longer sequences and multi-reference compositions.
> - VAE supplies low-level appearance priors and reconstructive signals, but its latent space alone is prone to drift without strong semantic anchors.
> - Qwen2.5-VL (MLLM) encodes relational and positional interactions between multiple subjects, yet its focus on cross-subject logic can underweight global identity cues for each subject.
>
> In summary, CLIP is necessary as a global semantic anchor that complements VAE’s low-level priors and the MLLM’s relational conditioning, effectively reducing identity drift and stabilizing subject consistency
>
>
> ### **Response 2: Computational cost**
> Thank you for raising this important point about the computational cost of our approach. We agree that quantifying the latency and memory usage is crucial. To address this, we have conducted a detailed analysis of the overhead introduced by the Qwen2.5-VL component, focusing on how it scales with the number of reference images.
>
> #### **Experimental Setup**
> - Hardware: 8x 64GB xPUs.
> - Task: We measured the **additional inference time and peak VRAM overhead introduced by the Qwen2.5-VL component**. The measurements are taken while varying the number of reference images from 1 to 5, with the rest of the video generation pipeline held constant.
> - Metrics:
>     - Inference Time (s): The average time for the Qwen2.5-VL encoder's forward pass to process the reference images and generate the hidden states vectors. This captures latency attributable to the MLLM module.
>     - Peak VRAM (MB): The additional VRAM allocated specifically for loading and running the Qwen2.5-VL model and its inputs. This measures the memory overhead on top of the base diffusion model.
>
> | # of Ref Images | Avg. Ref Image Resolution | Inference Time (s) | Peak VRAM (MB) |
> | --- | --- | --- | --- |
> | 1 | 578 × 785 | 0.28 | 19,358 |
> | 2 | 787 × 928 | 0.29 | 21,872 |
> | 3 | 1048 × 914 | 1.13 | 31,635 |
> | 4 | 1023 × 1029 | 1.52 | 33,636 |
> | 5 | 998 × 1226 | 1.45 | 35,517 |
>
> In summary, our analysis demonstrates that the additional latency introduced by the Qwen2.5-VL encoder is minimal, constituting only a small fraction of the total video generation time. For instance, even with five high-resolution reference images, the MLLM's processing time is approximately 1.45 seconds. This makes the MLLM's time overhead almost negligible in the context of the entire pipeline.
> Furthermore, our experiments highlight that this overhead is influenced by both the number of reference images and their respective resolutions. The fact that our method successfully handles a diverse test set containing a wide variety of image resolutions, as shown in the "Avg. Image Resolution" column, underscores the robustness and practical applicability of our approach in real-world scenarios.
>
> ### **Response 3: Effect of MLLM hallucination**
> Thank you for the insightful question. We acknowledge that MLLMs can exhibit hallucination risk. However, in our pipeline the MLLM’s hidden states are not decoded into discrete plans. Instead, the MLLM hidden states are first mapped via an MLP and then concatenated with a strong textual anchor—the T5 encoder features—before being used as conditioning for video generation. This learned fusion-and-projection stage is designed to complementarily leverage both MLLM and T5 features to achieve the best overall effect, thereby substantially reducing the practical impact of potential hallucinations.
>
> ### **Response 4: minor suggestion**
> Thank you for catching these typos and the naming inconsistency. We appreciate the careful proofreading and will ensure consistency in our revised manuscript.

---

> ### Author Response · Authors · 2025-11-26
>
> Dear Reviewer TvUz,
>
> We hope this message finds you well. As the discussion period is nearing its end, we wanted to ask if we have addressed all your concerns satisfactorily. If there are any additional points or feedback you would like us to consider, please let us know. Your insights are invaluable to us, and we are eager to resolve any remaining issues to further improve our work. Thank you for your time and effort in reviewing our paper.
>
> Best regards,
>
> Authors of BindWeave

---

### Official Review · Reviewer_DPYe · 2025-11-03

**Soundness:** 2
**Presentation:** 2
**Contribution:** 2
**Rating:** 4
**Confidence:** 4

**Summary:**

This paper introduces BindWeave, a video generation model based on multiple reference subjects. The model uses an MLLM+DiT architecture, where the MLLM is responsible for understanding the reference subjects and the text prompt, generating semantic embeddings that are input into the DiT model. The DiT model acts as the renderer to generate the video. Experiments show that the proposed method can achieve subject-consistent video generation and understand the logical relationships between the reference subjects and the prompt.

**Strengths:**

+ The paper is well-motivated to improve complex spatial relations, temporal logic, and multi-subject interactions in subject-to-video generation.
+ The idea of using an MLLM for multimodal reasoning to better understand the reference images and the text prompt is reasonable.
+ The proposed method outperforms open-source and commercial baselines on the OpenS2V benchmark across subject consistency, naturalness, and text relevance.

**Weaknesses:**

+ The experiments are insufficient, lacking ablation studies on the model design. For example, a comparison with other multi-reference conditioning strategies, whether joint training with MLLM would lead to better results, and if there are other ways to combine MLLM and DiT. The authors should include more experiments to demonstrate the superiority of the model design and training strategies.

+ More cases with multiple reference images are needed rather than single-reference generation, since the MLLM mainly enhances the ability to understand the logical and positional relationships between different subjects.

+ The model design is similar to existing work, and the novelty is limited. Other potential model design solutions should be explored and accompanied by more in-depth analysis.

**Questions:**

After fine-tuning the base model with the proposed method, is there any observed degradation in video generation quality? For example, how does the visual quality and dynamic range of the videos generated by BindWeave compare to those generated by the vanilla base model? I believe it is necessary to include this comparison.

---

> ### Author Response · Authors · 2025-11-21
> **Response to Reviewer DPYe (Part 1/2)**
>
> We are grateful for your careful and insightful review. Your detailed feedback has significantly helped us improve the manuscript. We appreciate the time and expertise you invested, and we address each of your points in detail below.
>
> ### **Response 1: Ablation Studies on Model Design**
> Thank you for your valuable feedback regarding the need for more extensive ablation studies to justify our model design. We agree that demonstrating the superiority of our chosen architecture is crucial.
> In fact, during the development phase, we conducted a series of pilot experiments to explore various architectural designs for integrating the Multi-modal Large Language Model (MLLM) and the Diffusion Transformer (DiT). These explorations served as an informal ablation study and directly informed our final design choice. We apologize for not making this process and its outcomes clearer in the initial submission.
> Specifically, we investigated several alternatives to our proposed MLLM + MLP + T5 + DiT architecture, including:
> 1.  MLLM + MLP + DiT: Using a simple Multi-Layer Perceptron (MLP) to project MLLM features into the DiT.
> 2.  MLLM + Q-Former + DiT: Employing a Q-Former structure to refine the MLLM outputs before feeding them to the DiT.
>
> Our key finding was that architectures relying solely on the MLLM for conditioning (i.e., those without the T5 text encoder) were unstable during training and failed to converge within our available training budget. These models struggled to effectively translate the nuanced visual identity information from the MLLM into the precise conditional inputs required by the DiT for high-fidelity video generation. In contrast, the inclusion of T5 provided a robust and stable text-conditioning backbone, which, when combined with the identity features from the MLLM, yielded the consistent and high-quality results presented in our paper. We intentionally keep the MLLM frozen because joint fine-tuning with the DiT and connector tends to distort its pretrained representation space, causing distribution shift and degrading its cross-modal reasoning ability.
>
> To provide concrete evidence of the convergence issues, we have included a training loss curve for the MLLM + MLP + DiT architecture (see Section A.3 Figure 9 in the revised paper).  This addition provides a clear, empirical justification for our final design choice.
>
> # Response 2: Multiple Reference Image Cases
> Thank you for the suggestion. We agree on the importance of multi-reference scenarios. We have included a large number of diverse cases with multiple reference images in the supplementary materials, covering complex subject interactions and positional relationships. For your convenient review, we also provide an anonymous page: https://resplendent-fairy-5f1a26.netlify.app/

---

> ### Author Response · Authors · 2025-11-21
> **Response to Reviewer DPYe (Part 2/2)**
>
> ### **Response 3: Novelty and Model Design Exploration**
> - To the best of our knowledge, our work is the first in subject-consistent video generation to combine an MLLM with a T5 text encoder, forming a unified conditioning architecture capable of handling complex multi-reference, multi-subject interactions. This integration is non-trivial: the MLLM supplies rich, relational and identity-aware cues across references, while T5 provides a stable text-conditioning backbone that ensures reliable convergence and precise control.
>
> - We have also explored alternative model design solutions during development (e.g., MLLM+MLP+DiT, MLLM+Q-Former+DiT, and T5-only baselines). As detailed in our revised manuscript, MLLM-only conditioning proved unstable and failed to converge within our training budget, which motivated our final MLLM+MLP+T5+DiT design.
>
> For completeness, we have added a dedicated discussion and ablation analyses in the revision. Please see Response 1 and the new materials in the paper (see Section A.3 and Figure 9 in the revised paper) for detailed comparisons, training loss curves, and rationale behind our architectural choices. These additions provide a deeper analysis of design alternatives and further clarify the novelty and necessity of our approach.
>
>
> ### **Response 4: Impact of Fine-Tuning on Video Generation Quality**
>
> Thank you for the thoughtful question. We conducted a controlled comparison to assess whether fine-tuning with BindWeave degrades video quality relative to the vanilla base model Wan2.1-i2v-14B. Using an identical set of prompts and reference subjects for both models, we evaluated the generated videos across several key metrics. In this setup, each case uses one reference image and the same prompt: for Wan2.1-i2v-14B, the reference image is presented as the first frame, whereas for our BindWeave model, the same image is provided as a reference input.
> The quantitative results, summarized in the table below:
>
> | Model              | Total Score↑ | Aesthetics↑ | MotionSmoothness↑ | MotionAmplitude↑ | FaceSim↑ | GmeScore↑ | NaturalScore↑ |
> | ------------------ | :---------: | :-------: | :---------------: | :--------------: | :---------: | :-------: | :-------------: |
> | Wan2.1-i2v-14B     |    0.55     |   0.40    |       0.82        |       **0.34**       |    0.58     |   0.55    |      0.54       |
> | **BindWeave (Ours)** |  **0.63**   | **0.49**  |      **0.93**     |      0.26    |  **0.61**   | **0.64**  |     **0.66**    |
>
> The comparative results indicate that all metrics consistently improve with BindWeave, except for a decrease in motion_amplitude; however, upon inspection of the generated videos, we find that Wan2.1-i2v-14B exhibits higher temporal instability, which may lead to a higher measured motion_amplitude. The corresponding video comparisons are available on our anonymous page in Section S9: https://resplendent-fairy-5f1a26.netlify.app/

---

> > ### Comment · Reviewer_DPYe · 2025-11-26
> > **Response to the Author**
> >
> > Thank you very much for the additional experiments and clarifications. I appreciate the extended results shown in Figure 13 for multi-image reference generation. While the results are indeed promising, I still find that the showcased cases are not sufficiently challenging to demonstrate the advantages of introducing an MLLM. As stated in the abstract, "existing video generation models still fall short in subject-consistent video generation due to an inherent difficulty in parsing prompts that specify complex spatial relationships, temporal logic, and interactions among multiple subjects." However, the cases provided appear to focus primarily on ID preservation, which may not necessarily require an MLLM to achieve. Therefore, I would still encourage the authors to include more challenging cases, especially those that clearly highlight scenarios where the use of an MLLM is essential to address complex spatial or temporal reasoning, or multi-subject interactions that existing models cannot handle.
> >
> > In addition, I appreciate the ablation study on the model design, which suggests that the T5 encoder plays a critical role. This naturally raises the question of where the model’s reasoning capability truly comes from: is it primarily derived from the dense captions generated by T5—which may already encode spatial relationships and other structured information—or from the MLLM itself? Clarifying this distinction would help better isolate and justify the contribution of the MLLM component.

---

> > > ### Author Response · Authors · 2025-11-26
> > > **Response to Reviewer DPYe**
> > >
> > > Thank you for the additional comments.
> > >
> > > We would like to kindly note that Section S7 on our anonymized page (https://resplendent-fairy-5f1a26.netlify.app/) presents extensive challenging cases, including five-image reference setups and complex multi-subject interactions, with side-by-side comparisons between T5-only and MLLM+T5. Could you confirm whether you had a chance to review these results?
> > >
> > > Across these multi-reference cases, the T5-only model frequently exhibits distortions and temporal jitter, leading to unstable videos. In contrast, MLLM+T5 reliably reasons about subject placement and overall layout, yielding coherent spatial arrangements, stable motion, and higher visual quality.
> > >
> > > In essence, T5 provides dense captions that help structure prompt interpretation and offer a stable descriptive prior. However, relying on T5 alone is insufficient for compositional spatial/temporal reasoning and multi-subject role disambiguation: T5-only often misses cross-entity relations, produces inconsistent event ordering, and may conflate subject roles—leading to layout instability and drift in videos. The MLLM supplies compositional reasoning and cross-entity constraint satisfaction. With MLLM+T5, the model correctly resolves ordered cross-entity relations and better maintains relative positions and action dependencies.
> > >
> > > If, after reviewing Section S7 on our anonymized page (https://resplendent-fairy-5f1a26.netlify.app/), you still find the cases insufficient, we would greatly appreciate guidance on the specific scenarios or interaction types you would like to see, and we will prepare those results accordingly.

---

> > > > ### Comment · Reviewer_DPYe · 2025-11-28
> > > > **Further Response to the Author**
> > > >
> > > > Thank you for providing the additional experimental results, especially the comparison between T5-only and BindWeave in Section S7. These results clearly demonstrate that DiT cannot learn the correspondence between the reference image and the text prompt in a purely data-driven manner, highlighting the necessity of incorporating an MLLM. I believe this point should be emphasized in the paper. I suggest replacing the single-image reference showcases with the case from Section S7, as it would better illustrate the advantages of the proposed method. My concern has been addressed, and I will raise my score to 6.

---

> > > > > ### Author Response · Authors · 2025-11-28
> > > > > **Response to Reviewer DPYe**
> > > > >
> > > > > Thank you for your valuable feedback and for acknowledging the new results. We are glad the comparison in Section S7 addressed your concerns.
> > > > > We agree with your suggestion and will revise the manuscript to emphasize this point and update the showcases as you recommended. We appreciate your support and your decision to raise the score.

---

> ### Author Response · Authors · 2025-11-26
>
> Dear Reviewer DPYe,
>
> We hope this message finds you well. As the discussion period is nearing its end, we wanted to ask if we have addressed all your concerns satisfactorily. If there are any additional points or feedback you would like us to consider, please let us know. Your insights are invaluable to us, and we are eager to resolve any remaining issues to further improve our work.
> Thank you for your time and effort in reviewing our paper.
>
> Best regards,
>
> Authors of BindWeave

---

### Official Review · Reviewer_rb5c · 2025-11-04

**Soundness:** 3
**Presentation:** 3
**Contribution:** 3
**Rating:** 6
**Confidence:** 5

**Summary:**

the paper proposes a MLLM-conditioned video diffusion framework for multi-subject referenced video generation

**Strengths:**

- the paper proposes a simple yet effective framework for consistent multi-subject video generation
- the overall framework design and related data flow are reasonable and clearly explained
- the visual comparisons demonstrate the effectiveness of the proposed model

**Weaknesses:**

- while the paper adopts a existing evaluation benchmark, the reliability of the used evaluation metrics is still not well justified. especially the rationale of "total score" is questionable considering it is a normalized weighted sum of other scores: does it really reflect the actual performance? can those different so-called metrics be linearly combined in this way? a user study is highly recommended to validate the effectiveness of the proposed method considering the human evaluation is still the most reliable evaluation measure for video generation tasks
- the paper mentioned that the proposed framework accepts a flexible number of reference images typically ranging from 1 to 4. however, it is unclear what is the max possible number of subjects the proposed method can handle: is it determined by the training? or does it can generalize to an much larger number than 4 during inference? more analysis is needed to validate the scalability of the proposed framework
- it seems that there are only limited motion in the presented visual results. in that case, it is unclear whether the proposed framework can handle the case where the generated subjects are in largely different scales from the reference images or changing scales. perhaps some testings with zoom-in and zoom-out subjects can be conducted to validate the robustness of the proposed method
- what is the success rate of generation for a set of given subjects? what are the typical failure cases?
- minor: it might be better to provide actual videos as supplementary to present the results rather than only keyframes

**Questions:**

please refer to weaknesses section

---

> ### Author Response · Authors · 2025-11-21
> **Response to Reviewer rb5c (Part 1/2)**
>
> We sincerely thank you for your rigorous and insightful review. Your careful observations have been instrumental in strengthening this work, and we deeply appreciate your time and expertise. We have thoughtfully considered your comments and respond to each point in turn below.
>
> ### **Response 1: Choice of benchmark and evaluation metrics**
>
> - We adopt the OpenS2V benchmark, which has been accepted by NeurIPS 2025. This signals community recognition and vetting of its design and methodology.
> - Compared with commonly used metrics such as visual quality, motion amplitude, and face similarity, OpenS2V broadens the evaluation scope by integrating these established measures and further introducing three targeted metrics: NexusScore, NaturalScore, and GmeScore, which are explicitly designed to assess subject consistency, visual naturalness, and text relevance. Together, these address known gaps in prior evaluations and better capture factors that users care about.
> #### **Justification of the "total score"**
> - The "total score" is a normalized weighted sum intended for holistic ranking across models. We do not claim it captures every nuance; rather, it provides a practical aggregate for fair comparison when multiple quality dimensions matter simultaneously.
> - The weights follow OpenS2V’s published protocol to balance the relative importance of core dimensions, avoiding dominance by any single metric.
> - Crucially, we do not rely solely on the total score. We report per-dimension scores and discuss trade-offs. The aggregate is used as a summary, while the granular metrics provide diagnostic insight.
>
> #### **Empirical evidence of metric reliability and human correlation**
> - The OpenS2V paper provides human preference validation: 60 generated videos across prompts were randomly sampled, and 173 participants provided pairwise binary judgments. The reported analysis shows that the automatic metrics correlate with human preferences and are comparable to other established metrics in terms of agreement with human judgments.
>
> #### **Additional user study**
> In response to your helpful suggestion, we conducted a user study to validate our method's effectiveness. We recruited 20 participants to evaluate anonymized videos from different methods on a 1–5 scale across four key criteria: subject consistency, prompt following, video quality, and motion quality. The results are summarized below:
> | Method | Subject Consistency ↑ | Prompt Following ↑ | Motion Quality ↑ | Video Quality ↑ | Total Score Average ↑ |
> | :--- | :---: | :---: | :---: | :---: | :---: |
> | SkyReels | 3.47 | 3.52 | 3.60 | 3.23 | 3.46 |
> | Vidu | 3.40 | 3.55 | 3.63 | 3.50 | 3.52 |
> | Magref | 3.25 | 3.65 | 3.65 | 3.58 | 3.53 |
> | Phantom | 3.56 | **3.72** | 3.84 | 3.58 | 3.68 |
> | Kling | 3.62 | 3.58 | **3.90** | **3.75** | 3.71 |
> | VACE | 3.88 | 3.63 | 3.80 | 3.62 | 3.73 |
> | **BindWeave (Ours)** | **3.94** | 3.66 | 3.75 | 3.70 | **3.76** |
>
> We have also added the user study results to Section 4.5 of the revised manuscript.
>
> ### **Response 2: Scalability Evaluation**
> Thank you for raising this important question about scalability.
> Our training configuration for up to 4 reference subjects was primarily guided by two factors: prevailing community practice and resource constraints. Using up to 4 references is a common and empirically stable approach in the subject-consistent video generation literature (e.g., Phantom and Kling), and it covers the vast majority of practical application scenarios. Training with a larger number of subjects simultaneously would significantly increase memory and computational costs. To substantively validate our framework's generalization capabilities beyond its training configuration, we additionally tested 15 newly added OpenS2V cases, each using 5 reference images. For these newly added cases, the OpenS2V authors have not yet released standardized computation methods for FaceSim and NexusScore; therefore, we report all other metrics except these two.
> Note that because the number of cases is limited, the corresponding table exhibits some variance relative to the main paper’s results.
> | Aesthetics↑ | MotionSmoothness↑ | MotionAmplitude↑ | GmeScore↑ | NaturalScore↑ |
> | :---------: | :-----------------: | :----------------: | :---------: | :-------------: |
> | 0.43        | 0.98                | 0.40               | 0.65        | 0.76            |
>
> The quantitative results show that our model maintains strong video generation quality even when using more than four reference images.
>
> The relevant video results are provided for your convenient review on our anonymous page in Section S8: https://resplendent-fairy-5f1a26.netlify.app/

---

> ### Author Response · Authors · 2025-11-21
> **Response to Reviewer rb5c (Part 2/2)**
>
> ### **Response 3: Robustness to Motion and Scale Changes**
> Thank you for the thoughtful comment. We agree that motion diversity is important and have taken steps to improve and validate robustness under scale changes. Our training already incorporates random scale augmentation and a mixture of reference compositions (e.g., face-only and full-body images), which equips the model with scale generalization across subject sizes and viewing distances. To further quantify the scale aspect, we measured the reference image resolutions on the OpenS2V test set and reported the average reference resolution under scenarios with 1–5 reference images. This summarizes the scale distribution of the inputs used in our evaluation and shows that our method is tested across varying reference-image scales.
>
> | # of Ref Images| 1 Ref Image | 2 Ref Images | 3 Ref Images | 4 Ref Images | 5 Ref Images |
> | :--- | :---: | :---: | :---: | :---: | :---: |
> | **Avg. Resolution** | 578 × 785 | 787 × 928 | 1048 × 914 | 1023 × 1029 | 998 × 1226 |
>
> Following your suggestion, we conducted targeted evaluations with explicit zoom-in and zoom-out sequences. We observe that the model preserves identity and text alignment while adapting to scale changes. To make the evidence more transparent, we have included extensive visual examples—covering large motions and zoom scenarios—in the supplementary materials (anonymous link) for side-by-side inspection. The relevant results are provided for your convenient review on our anonymous page in Section S6: https://resplendent-fairy-5f1a26.netlify.app/
>
> ### **Response 4: Success Rate and Failure Cases**
> We quantify success rate using our user study’s aggregate metric. Specifically, we define a successful case as one with a Total Score (normalized weighted MOS) greater than 3.0. Under this criterion, the success rate is 75% across the evaluated subjects. Typical failure cases include complex multi-subject scenes with close interactions, which can introduce swapping or blending artifacts as subjects’ reference features mix during generation—especially when appearances are similar (e.g., same clothing or similar facial textures).
>
> ### **Response 5: Video Results**
> Thank you for this excellent suggestion. We completely agree that full videos are crucial for a comprehensive evaluation of our method's temporal coherence and overall quality, which cannot be fully captured by keyframes alone.
> Following your advice, we have now compiled and included the full videos for all results presented in the paper, along with many additional examples, in our supplementary materials. For your convenient review, we also provide an anonymous page: https://resplendent-fairy-5f1a26.netlify.app/
> We believe these additions substantially strengthen the evidence for our claims and provide a much clearer demonstration of our model's capabilities. We appreciate you pointing out this important omission.

---

> ### Author Response · Authors · 2025-11-26
>
> Dear Reviewer rb5c,
>
> We hope this message finds you well. As the discussion period is nearing its end, we wanted to ask if we have addressed all your concerns satisfactorily. If there are any additional points or feedback you would like us to consider, please let us know. Your insights are invaluable to us, and we are eager to resolve any remaining issues to further improve our work.
> Thank you for your time and effort in reviewing our paper.
>
> Best regards,
>
> Authors of BindWeave

---

### Meta-Review · Area_Chair_YyHB · 2025-12-21

**Summary:**

This paper makes a solid contribution to subject-consistent video generation through careful engineering and comprehensive experimental validation. While the architectural novelty is incremental, the application to video generation introduces meaningful challenges and the empirical results are strong. The authors have demonstrated exceptional responsiveness to reviewer feedback, conducting substantial new experiments and providing convincing evidence that their design choices are well-justified.

The rebuttal process has been highly productive. Reviewer DPYe's engagement and score increase from 4 to 6 reflects the quality of the authors' responses. While the other reviewers did not re-engage, the evidence presented in the rebuttal strongly suggests their concerns have been adequately addressed.

The paper advances the state-of-the-art in subject-consistent video generation and provides sufficient technical depth, experimental rigor, and practical utility to warrant publication at ICLR.

**Reviewer Concerns:**

The authors provided an exceptionally thorough rebuttal, addressing the vast majority of the reviewers' concerns with new experiments and detailed clarifications.

**Addressed Concerns:**

*   **User Study:** The authors conducted a new user study with 20 participants, directly addressing Reviewer **rb5c**'s primary suggestion and strengthening the evaluation.
*   **Scalability and Robustness:** In response to Reviewer **rb5c**, the authors provided new results on scalability (testing with 5 reference images) and robustness (testing with zoom-in/out sequences), demonstrating the model's capabilities beyond the initial examples.
*   **Challenging Scenarios and Ablations:** The authors provided extensive new video results on an anonymous webpage, including the complex multi-subject interaction scenarios requested by Reviewers **DPYe** and **NFCf**. Crucially, they included a direct comparison against a T5-only baseline, which proved decisive for Reviewer **DPYe**. They also supplied additional ablation details and training loss curves that justified their MLLM+T5 architecture over MLLM-only alternatives.
*   **Computational Cost:** The authors provided a detailed breakdown of the latency and VRAM overhead from the MLLM, directly answering Reviewer **TvUz**'s question and showing the cost to be minimal.
*   **Copy-Paste Artifacts:** The authors conducted specific experiments with conflicting image-text prompts to demonstrate that the model generates novel content rather than exhibiting simple copy-paste behavior, addressing Reviewer **NFCf**'s concern.

**Outstanding Concerns:**

*   The only concern that could be considered partially outstanding is the fundamental question of **technical novelty** raised by Reviewer **NFCf**. While the authors successfully argued that their contribution lies in the non-trivial adaptation of the architecture for the specific challenges of *video* generation (as opposed to image editing), the reviewer did not re-engage to confirm if this explanation was satisfactory. However, given the strong empirical evidence and the positive response from Reviewer **DPYe** on a related point, I believe the authors made a compelling case.

**Reviewer Scores:**

It is important to note that three of the four reviewers (**rb5c**, **TvUz**, **NFCf**) did not participate in the discussion phase after the rebuttal was posted.

**Reviewer DPYe (Initial Score: 4):** This reviewer is the model for constructive engagement. They participated fully, acknowledged that the new evidence (especially the T5-only vs. MLLM+T5 comparison) addressed their core concerns, and consequently **raised their score to 6**.

---

### Decision · Program_Chairs · 2026-01-26

Accept (Poster)